EMBO
Molecular Medicine

# Lymphopenia drives T cell exhaustion in immunodeficient STING gain-of-function mice

Damien Freytag[1], Stéphane Giorgiutti [1,2,3], Grégoire Hopsomer [1], Nadège Wadier [1,3], Sabine Depauw[1,3], Philippe Mertz [1], Fabrice Augé [4], Raphaël Carapito [1], Isabelle Couillin [5], Anne-Sophie Korganow[2,3,11], Francesca Pala [6], Marita Bosticardo [6], Luigi D Notarangelo[6], Frédéric Rieux-Laucat [7], Nicolas Riteau [5,8], Peggy Kirstetter [9] & Pauline Soulas-Sprauel [1,2,10] ✉

## Abstract

STING gain-of-function (GOF) mutations cause STING-Associated Vasculopathy with onset in Infancy (SAVI), a severe autoinflammatory disease. Mice carrying STING GOF V154M mutation develop profound T cell lymphopenia, partly due to impaired thymic development. To investigate the mechanisms of peripheral T cell dysfunctions, we analyzed transcriptomic and phenotypic profiles of splenic T cells from these mice. We found a terminally exhausted T cell phenotype, established early in life upon entry into the periphery, independent of type I interferons and intrinsic STING activation in T cells or stromal cells. Mechanistically, naïve T cells in the lymphopenic periphery experienced heightened stimulation of the IL-7 receptor and TCR, including NFAT pathway, a key factor in T cell exhaustion. Transplantation of STING GOF hematopoietic stem cells with wild-type bone marrow prevented exhaustion in this non-lymphopenic context, placing lymphopenia as a key driver. T cell exhaustion was also observed in lymphopenic mice carrying *Rag1* hypomorphic mutations. In conclusion, our results highlight T cell exhaustion induced by lymphopenia and could have important implications for the management of patients with severe immune deficiencies.

Keywords STING; T Cell Exhaustion; Lymphopenia; Homeostasis
Subject Category Immunology

## Introduction

Stimulator of interferon genes (STING) belongs to cytosolic double-stranded-DNA-sensing pathways in innate immunity and has emerged as a central player in antiviral, antibacterial, and antitumor immunity (Ablasser and Chen, 2019). When activated by cyclic dinucleotides produced by cyclic GMP-AMP synthase (cGAS), STING translocates from the endoplasmic reticulum (ER) to the ER-Golgi intermediate compartment (ERGIC)/Golgi, where it activates TANK-binding kinase 1 (TBK1), which then phosphorylates interferon regulatory factor 3 (IRF3) and induces the expression of type I interferons (IFNs). In turn, type I IFNs activate the type I IFNs receptor (IFNAR), inducing the expression of a set of interferon-stimulated genes (ISG) and triggering inflammation. The NF-kB pathway is also engaged following STING activation, participating in the inflammatory response (Hopfner and Hornung, 2020).

Heterozygous gain-of-function (GOF) mutations in STING-coding gene *STING1* lead to constitutive activation of the protein and have been described in patients with an autoinflammatory disease designated STING-associated vasculopathy with onset in Infancy (SAVI) (Liu et al, 2014; Jeremiah et al, 2014), classified as a type I interferonopathy, with increased ISGs expression (Crow and Stetson, 2021). This disease is associated with damages to the skin small blood vessels and an interstitial lung disease at the origin of morbidity and mortality (Frémond et al, 2021; Frémond and Crow, 2021; David and Frémond, 2022). To study the role of STING in the pathophysiology of the disease, our team (Bouis et al, 2019) and others (Motwani et al, 2019; Warner et al, 2017; Luksch et al, 2019; Bennion et al, 2019; Martin et al, 2019; Siedel et al, 2020; Bennion et al, 2020) developed STING GOF mouse models including mice carrying the heterozygous V154M mutation (Bouis et al, 2019; Motwani et al, 2019), corresponding to the most common human V155M mutation

[1]Laboratoire d'ImmunoRhumatologie Moléculaire, Plateforme GENOMAX, INSERM UMR_S 1109, Faculté de Médecine, Fédération Hospitalo-Universitaire OMICARE, ITI TRANSPLANTEX NG, Strasbourg Federation of Translational Medicine (FMTS), Strasbourg University, Strasbourg, France. [2]Department of Clinical Immunology and Internal Medicine, National Reference Center for Systemic Autoimmune Diseases (CNR RESO), Tertiary Center for Primary Immunodeficiency, Strasbourg University Hospital, F-67000 Strasbourg, France. [3]University of Strasbourg, Faculty of Medicine, Strasbourg, France. [4]IBMC, Institut de Biologie Moléculaire et Cellulaire, F-67000 Strasbourg, France. [5]Experimental and Molecular Immunology and Neurogenetics Laboratory, University of Orleans, Centre National de la Recherche Scientifique (CNRS), UMR7355, 45100 Orleans, France. [6]Laboratory of Clinical Immunology and Microbiology, National Institute of Allergy and Infectious Diseases (NIAID)/NIH, Bethesda, MD, USA. [7]Université Paris Cité, Laboratoire d'immunogénétique des maladies auto-immunes pédiatriques, Institut Imagine, INSERM UMR_S1163, Paris, France. [8]Immune Health Laboratory, 'Regulation of host responses and immune health' IRL2029, French National Centre for Scientific Research (CNRS) and Ribeirão Preto Medical School (FMRP) of the São Paulo University (USP), São Paulo, Brazil. [9]Université de Strasbourg, IGBMC UMR 7104- UMR-S 1258, F-67400 Illkirch, France; IGBMC, Institut de Génétique et de Biologie Moléculaire et Cellulaire, F-67400 Illkirch, France. [10]University of Strasbourg, Faculty of Pharmacy, Illkirch, France. [11]Present address: Laboratoire d'ImmunoRhumatologie Moléculaire, Plateforme GENOMAX, INSERM UMR_S 1109, Faculté de Médecine, Fédération Hospitalo-Universitaire OMICARE, ITI TRANSPLANTEX NG, Strasbourg Federation of Translational Medicine (FMTS), Strasbourg University, Strasbourg, France. ✉E-mail: soulaspa@unistra.fr

(about 60%) in SAVI patients. STING GOF V154M mouse models develop type I IFN-independent autoinflammatory manifestations, notably lung disease (Motwani et al, 2019; Gao et al, 2022), and also a high decrease of T cell counts in a context of severe combined immunodeficiency (SCID) (Bouis et al, 2019; Motwani et al, 2019), reminiscent of T cell lymphopenia cases among SAVI patients. T cell lymphopenia was also described in STING GOF N153S models (Motwani et al, 2019; Warner et al, 2017). In STING GOF N153S and V154M mice, the T cell lymphopenia is partly explained by an impairment of T cell development at double negative (DN)1 and DN2 stages (Bouis et al, 2019; Warner et al, 2017).

In addition, we showed that the rare mature T cells found in the periphery exhibited both defective proliferation and increased apoptosis, which maintains the lymphopenic state (Bouis et al, 2019). These defects are at least partly STING-intrinsic as the anti-proliferative effect of STING was confirmed in human T cells overexpressing the STING GOF V155M mutant (Cerboni et al, 2017). In addition, a pro-apoptotic effect of STING in lymphocytes was also reported in other models, and involves TBK1 (Gulen et al, 2017), endoplasmic reticulum (ER) stress (Wu et al, 2019) or p53-dependent pathways (Concepcion et al, 2022). However, treatment of STING N153S mice with a pan-caspase inhibitor or ER stress blocker only partially reverses T cell lymphopenia (Wu et al, 2019). Thus, other mechanisms must explain the functional defects of peripheral T cells and maintenance of lymphopenia in STING GOF mice.

To better characterize these mechanisms, we perform in the present study a transcriptomic analysis on sorted T cells from STING GOF mice compared to their wild-type (WT) littermate controls. An enrichment of transcriptomic signatures associated with T cell exhaustion is revealed and confirmed by the overexpression of key transcription factors and several surface inhibitory receptors (like PD-1, TIGIT, TIM-3, and LAG-3). We further demonstrate that T cell exhaustion is associated with enhanced TCR and IL-7R stimulation in the lymphopenic context, impacting already T cells at the naive stage, before their transition to the exhausted state. Importantly, we do not observe any sign of T cell exhaustion in STING GOF T cells developing in non-lymphopenic mice. Finally, a similar T cell exhaustion phenotype (albeit less severe than in STING GOF mice) is observed in other lymphopenic conditions, as shown in mice carrying hypomorphic mutations of the *Rag1* gene (Ott De Bruin et al, 2018). Together, these observations demonstrate that the T cell exhaustion phenotype in STING V154M mice is mostly triggered by lymphopenia and likely reinforces this lymphopenia by a self-maintaining loop. While T cell exhaustion has primarily been described as a consequence of chronic T cell stimulation in the context of persistent infections and cancers (Wherry and Kurachi, 2015; McLane et al, 2019), our study reveals that lymphopenia should also be considered as a context that can trigger T cell exhaustion, echoing recent findings of T cell exhaustion in patients with severe combined immunodeficiencies (SCID) (Labrosse et al, 2023; Dong et al, 2023).

# Results

## CD4+ and CD8+ T cells from STING GOF mice display an exhausted phenotype, acquired in the periphery in a type I IFN-independent manner

Mice carrying the heterozygous STING V154M mutation are characterized by a SCID phenotype associated with marked T cell dysfunctions, including impaired proliferation, and enhanced apoptosis (Bouis et al, 2019). To decipher the mechanisms underlying this phenotype, we performed a transcriptomic analysis by RNA-seq on sorted splenic CD4+ and CD8+ T cells from STING GOF mice compared to their WT littermate controls. This analysis revealed a transcriptomic signature of T cell exhaustion in CD4+ and CD8+ T cells in the STING GOF group, as shown by a negative enrichment of the transcripts upregulated in naive versus exhausted T cells, as well as a positive enrichment of the genes downregulated in naive versus exhausted T cells (GSE9650 (Wherry et al, 2007)) (Fig. 1A). T cell exhaustion is an altered differentiation state of chronically activated T cells, notably caused by persistent antigenic stimulation and sustained inflammation, and commonly occurring in chronic viral infections or cancers (Wherry and Kurachi, 2015; McLane et al, 2019). Exhausted T cells co-express inhibitory receptors (like PD-1, TIGIT, TIM-3, and LAG-3) in a sustained manner (McLane et al, 2019). Furthermore, exhausted T cells progressively lose their proliferative and effector functions, which is reminiscent of the functional defects of mature STING GOF T cells (Bouis et al, 2019; Motwani et al, 2019).

Analysis of gene expression for key transcription factors associated with T cell exhaustion in STING GOF T cells revealed overexpression of *Nfatc1 and Nfatc2* compared to WT cells, and downregulation of the AP-1 cofactors *Fos* and *Jun* (Fig. 1B). This pattern is indicative of a preferential activation of the transcriptional program of exhaustion (Wherry et al, 2007; Martinez et al, 2015). We also observed increased expression of T cell exhaustion-associated transcription factors, like EOMES (*Eomes*) and T-BET (*Tbx21*), as well as TOX (*Tox*), the latter being more involved in the most advanced stages of the T cell exhaustion program (Fig. 1B). Moreover, transcripts of multiple genes encoding for inhibitory receptors (*Pdcd1*, *Tigit*, *Havcr2*, *Lag3*, *Cd244*, *Entpd1*, *Ctla4*, and *Cd160*) were also overexpressed (Fig. 1C). Accordingly, the percentages of splenic CD4+ and CD8+ T cells positive for PD-1, TIGIT, TIM-3 or LAG-3 expression were significantly increased in STING GOF mice compared to their WT controls (Fig. 1D). Then, we analyzed the expression of additional markers to define more precisely the exhaustion stage of T cells in STING GOF mice. As the *Tcf7* transcript was downregulated in our RNA-seq data (Fig. 1B), we hypothesized that STING GOF T cells could be terminally exhausted. We analyzed co-expression of the inhibitory receptors PD-1 and TIM-3, as well as the transcription factor TCF-1 (encoded by *Tcf7* gene), to identify progenitor exhausted (PE) T cells (PD-1+TCF-1+TIM-3-) and terminally exhausted (TE) T cells (PD-1+TCF-1-TIM-3+) (Guo et al, 2021) (Fig. 1E). An increased proportion of TE T cells was observed among CD4+ and CD8+ T cells from STING GOF mice as compared to control mice. Interestingly, the STING GOF T cell exhaustion phenotype appeared to be more profound in CD4+ than in CD8+ T cells (Fig. 1F). In addition, the exhausted state of STING GOF T cells was further validated by a decreased expression of *Ifng* (encoding IFN-γ) and *Gzmb* (encoding Granzyme B) transcripts in sorted exhausted (PD-1+ / TIM-3+) compared to non-exhausted (PD-1-/ TIM-3-) T cells (Fig. 1G,H). Finally, we analyzed the expression of inhibitory receptors on splenic naive and memory T cells, in addition to total T cells. This analysis (made on a new cohort of mice) confirmed the statistically significant increased expression of STING GOF total T cells (Fig. EV1A). Interestingly, while central memory (CD44highCD62L+) and effector memory (CD44+CD62L-) T cells were overexpressing all inhibitory receptors studied, this was

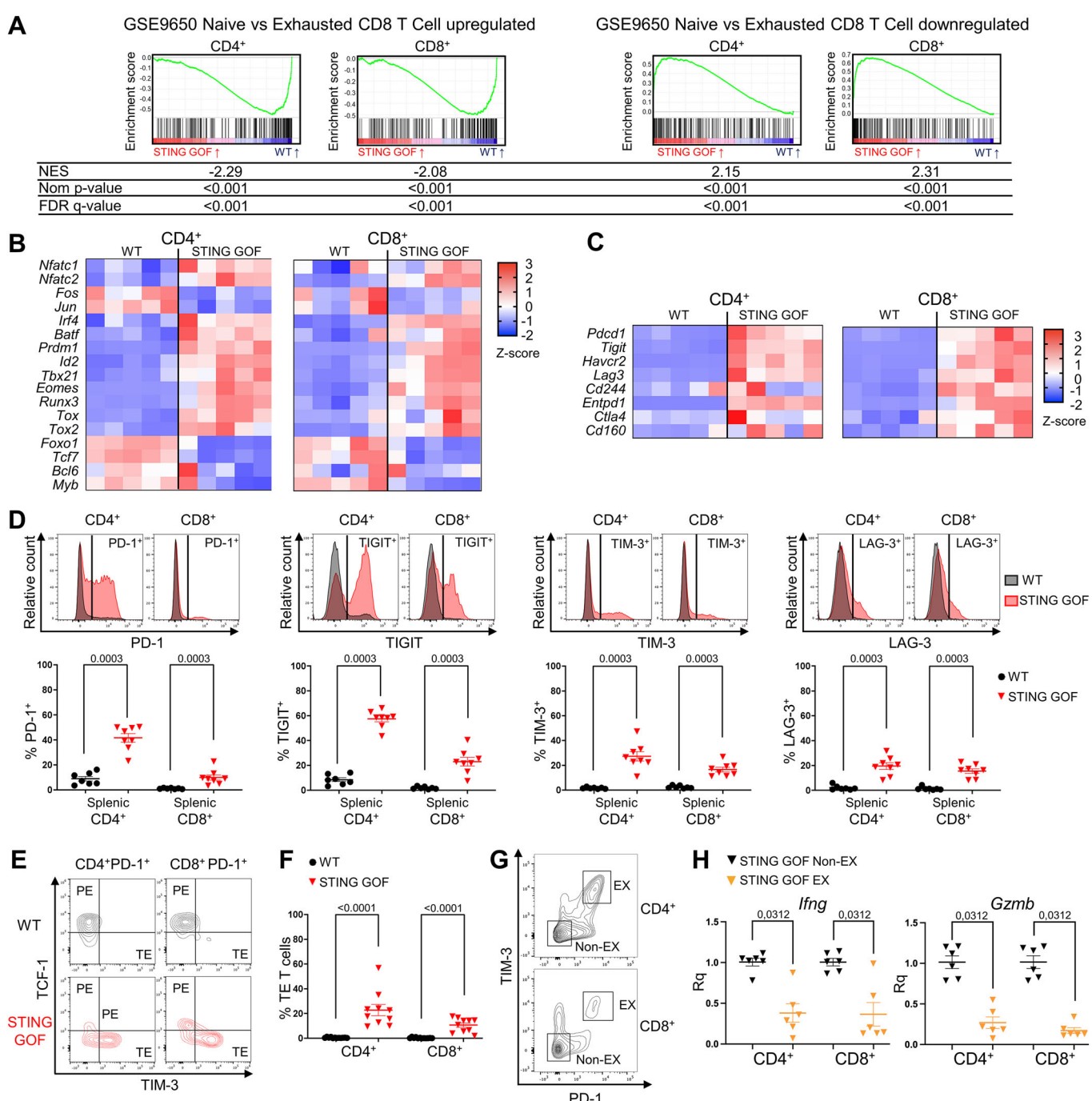

not the case for naive (CD44^low CD62L^+) T cells, which only presented a statistically significant and slight increase of TIGIT expression. In particular, the marker of terminal exhaustion TIM-3 was not increased on naive T cells (Fig. EV1B). Together, our findings revealed an unexpected phenotype of T cell exhaustion in STING GOF mice, which seems to be acquired during the transition from naive to memory T cells, reaches the stage of terminal exhaustion, and might contribute to the T cell dysfunctions described in these mice.

Given that STING V154M mice develop a lung disease partly mediated by T cells (Gao et al, 2022), we also investigated potential

T cell exhaustion in the lungs. Interestingly, as in the spleen, the proportion of CD4^+ and CD8^+ T cells infiltrating the lungs that expressed PD-1 (data available for CD4^+ T cells), TIGIT, TIM-3 or LAG-3 was significantly higher in STING GOF mice compared to WT controls (Fig. EV2A).

To determine if this exhaustion phenotype has a central origin, we examined whether the single positive (SP) T cells, the most terminally differentiated T cell progenitors in the thymus, were already exhibiting the phenotype. We observed only a minimal increase in cells expressing inhibitory receptors, for both thymic SP CD4^+ and CD8^+ progenitors (Fig. EV2B). In contrast, we

**Figure 1. Splenic CD4⁺ and CD8⁺ T cells from STING GOF mice display an exhausted phenotype.**

(A–C) Analysis of RNA-seq performed on sorted splenic CD4⁺ or CD8⁺ T cells from STING GOF mice ($n = 5$) and their WT littermate controls ($n = 5$). (A) Gene set enrichment analysis (GSEA) of the signatures of genes upregulated or downregulated in naive versus exhausted CD8⁺ T cells (GSE9650) among genes deregulated in STING GOF versus WT CD4⁺ or CD8⁺ T cells. Enrichment plots, normalized enrichment score (NES), nominal $p$ value (Nom $p$ value) and false discovery rate $q$ value (FDR $q$ value) are shown for each analysis. (B) Heatmap representing T cell exhaustion/dysregulated transcription factors mRNA expression (Z-score) by CD4⁺ or CD8⁺ T cells in STING GOF mice compared to WT mice. (C) Heatmap representing inhibitory receptors mRNA expression (Z-score) by CD4⁺ or CD8⁺ T cells in STING GOF and WT mice. (D–F) Immunophenotyping of splenic T cells from STING GOF mice and their WT littermate controls by flow cytometry. (D) Proportion of PD-1-, TIGIT-, TIM-3- and LAG-3-expressing cells among CD4⁺ or CD8⁺ T cells. Representative histograms are shown. (E) Contour plots showing progenitor exhausted (PE) and terminally exhausted (TE) T cells, thanks to TCF-1 and TIM-3 staining, gated on CD4⁺ PD-1⁺ and CD8⁺ PD-1⁺ T cells, respectively. (F) Proportion of terminally exhausted (TE) T cells among total CD4⁺ or CD8⁺ T cells. (G) Representative contour plots showing the gating strategy used to sort exhausted (EX; PD-1⁺ TIM-3⁺) and non-exhausted (Non-EX; PD-1⁻ TIM-3⁻) CD4⁺ or CD8⁺ T cells from the spleen of STING GOF mice. (H) Quantification by qPCR (comparative Ct method) of *Ifng* and *Gzmb* mRNA expression in sorted STING GOF EX versus Non-EX T cells. Data are normalized to the geometric mean of two housekeeping genes (*Actb* and *Gapdh*). Each data point corresponds to one mouse; mean ± SEM are shown per population for six to twelve mice from at least four experiments (biological replicates). Statistical significances are calculated with two-tailed Mann–Whitney test (D–F) or using paired two-tailed Wilcoxon test (H).

demonstrated that splenic T cells from very young (2-week-old) mice, which represent cells that have just left the thymus for the periphery, were already displaying the exhaustion phenotype of adult mice (Fig. EV2C). Altogether, these results demonstrate that the T cell exhaustion phenotype in STING GOF mice is acquired very early after birth, but only once T cells have reached the periphery, and gets worse from the naive stage to the memory stages.

Since STING drives type I IFNs, which have been shown to regulate coinhibitory receptors expression on T cells (Sumida et al, 2022), we then explored the role of these cytokines in T cell exhaustion in STING GOF mice. Overall, the T cell exhaustion phenotype was not reversed by IFNAR deficiency in STING GOF mice (Fig. EV3A,B).

## Ca²⁺-NFAT pathway is activated by STING, but is not sufficient to induce T cell exhaustion

We then searched for signals that could induce the exhaustion phenotype in peripheral STING GOF T cells. Considering its crucial role in T cell exhaustion (Seo et al, 2021) and its upregulation in our RNA-seq data (Fig. 1B), we first focused on the NFAT signaling pathway.

The transcriptomic analysis confirmed a significant enrichment of NFAT signature from the pathway interaction database (PID) in STING GOF T cells, compared to control T cells (Fig. EV4A). The upregulation of the total (nuclear and cytosolic) NFATc1 transcription factor expression was confirmed by flow cytometry (Fig. EV4B). NFAT activation relies on its dephosphorylation by the calcium (Ca²⁺)-calmodulin-calcineurin pathway, and Ca²⁺ homeostasis is known to be disrupted in STING GOF T cells (Wu et al, 2019). Using Fura Red™-mediated ratiometric measurement to study basal Ca²⁺ levels, we demonstrated that basal Ca²⁺ levels were higher in CD4⁺ and CD8⁺ T cells from STING GOF mice compared to their WT controls, suggesting that the Ca²⁺-NFAT signaling pathway in STING GOF T cells could be associated with the induction of their exhaustion (Fig. EV4C).

We next wondered whether the activation of the Ca²⁺-NFAT signaling pathway could be due to the STING activation per se. Therefore, we treated WT splenocytes with STING agonist 5,6-dimethylxanthenone-4-acetic acid (DMXAA) in vitro. After 4 h, DMXAA-treated CD4⁺ and CD8⁺ T cells presented an increase of their basal Ca²⁺ levels, similar to those observed in untreated

STING GOF T cells (Fig. 2A). After 24 h, DMXAA-treated cells also showed an upregulation of NFATc1 to the same extent as untreated STING GOF T cells (Fig. 2B). However, STING activation in vitro was not sufficient to induce an increase of key inhibitory receptors PD-1 and TIM-3 (Fig. 2C). It was not possible to perform longer stimulation times, given the mortality induced by the STING agonist.

In conclusion, even if it can contribute to the Ca²⁺-NFAT activation, STING activation in T cells seems insufficient to induce their exhaustion, reinforcing the importance of an environmental factor that was already suggested comparing thymic versus splenic exhaustion profile from 2-week-old STING GOF mice (Fig. EV2C).

## STING GOF T cells displayed features of TCR and IL-7R engagement mediated by the lymphopenic environment

We then hypothesized that NFAT activation in STING GOF could be the consequence of TCR signaling secondary to antigenic stimulations (Seo et al, 2021). Indeed, beyond NFAT overexpression, the transcriptomic analysis confirmed a significant enrichment of a broader TCR downstream pathway signature from the pathway interaction database (PID) in STING GOF T cells compared to control T cells (Fig. 3A). This was confirmed with three other GSEA signatures (Appendix Table S1).

In addition, we highlighted a significant downregulation of CD3 expression on the surface of STING GOF T cells compared to control T cells (Fig. 3B), coherently with the decreased response of STING GOF T cells to anti-CD3 stimulation described in our model (Bouis et al, 2019). As the downregulation of CD3 expression at the cell surface was not associated with a decrease of mRNA expression in our RNA-seq analysis (Appendix Table S2), it appeared to be more likely due to its internalization, consistent with TCR stimulation. Interestingly, the analysis of CD3 expression on splenic T cell subpopulations showed that CD3 expression is already significantly decreased at the naive stage and is thus not a consequence of T cell exhaustion (Fig. EV5A,B).

This antigenic stimulation was not secondary to an acute or chronic infection status, as STING GOF mice were bred and held under a specific opportunist-pathogen-free (SOPF) environment, and was not associated with cancer, as no tumors were detected in these mice. Thus, we hypothesized that increased TCR stimulation by conventional dendritic cells (cDCs) could be responsible for this

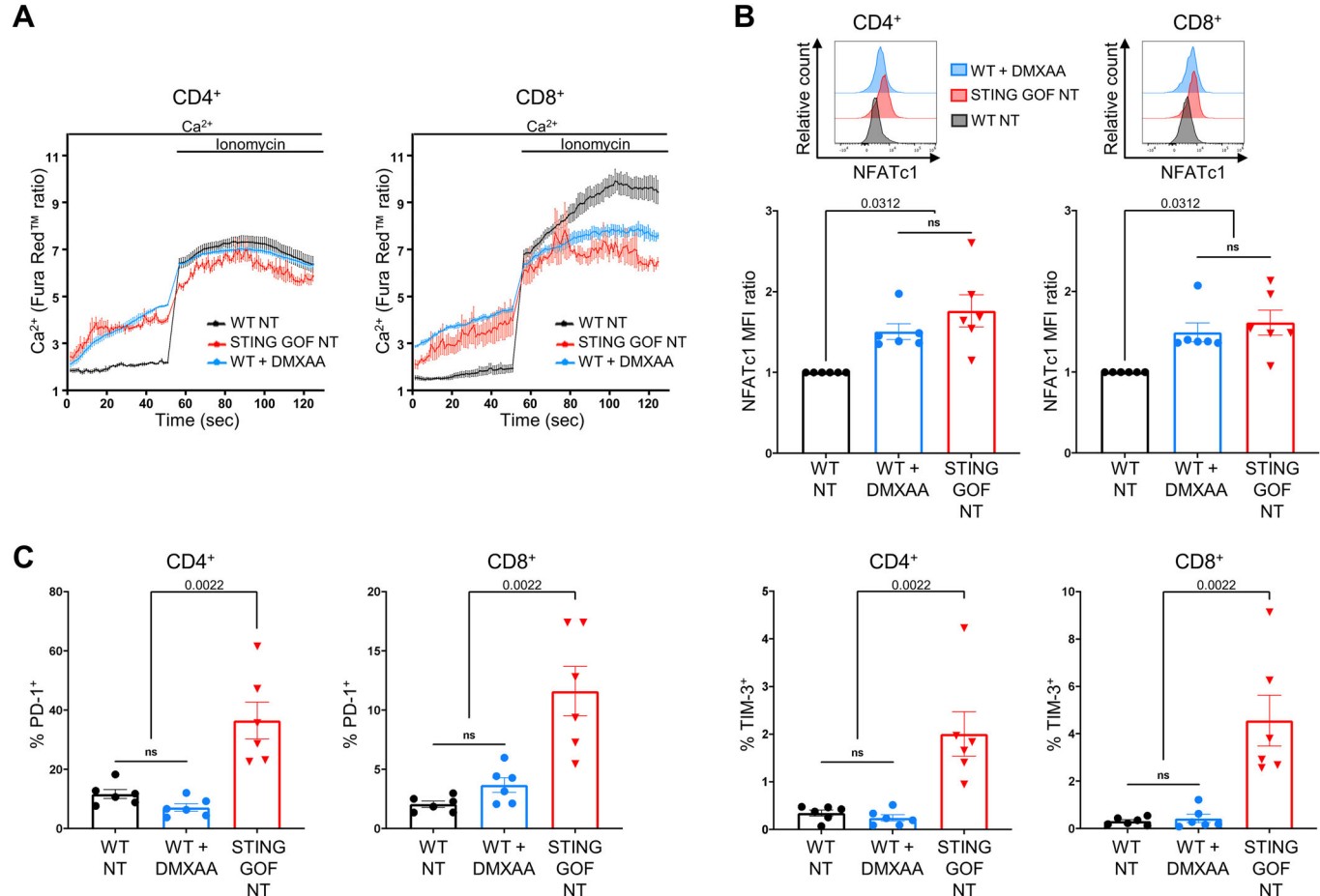

**Figure 2. Ca²⁺-NFAT pathway is activated by STING, but is not sufficient to induce T cell exhaustion.**

(A–C) Splenic WT splenocytes were treated in vitro with DMXAA (10 μg/mL) and compared to untreated (NT) WT and STING GOF splenocytes. After 4 h (calcium levels measurement) or 24 h (immunophenotyping), cells were analyzed by flow cytometry. (A) Relative cytosolic Ca²⁺ levels monitored by Fura Red™ ratio in CD4⁺ or CD8⁺ T cells. Splenocytes were treated, recorded, and plotted as in Fig. EV3C. Each data point represents the mean of three independent experiments, and their error bars the SEM. (B) Ratio of total NFATc1 MFI in CD4⁺ or CD8⁺ T cells. Ratio was normalized on the corresponding WT untreated (NT) control. Representative histograms are shown. (C) Proportion of PD-1- and TIM-3-expressing cells among CD4⁺ or CD8⁺ T cells. Each data point corresponds to one mouse; (B, C) Mean ± SEM are shown for six mice per population, from six independent experiments (biological replicates). For NFATc1 MFI ratio, statistical significances are calculated with Wilcoxon signed-rank test with a hypothetical value of 1. For PD-1⁺ and TIM-3⁺ proportions, statistical significances are calculated with a two-tailed Mann–Whitney test; ns (non-significant), $P > 0.05$. Source data are available online for this figure.

phenotype, similar to what is observed during homeostatic proliferation of T cells in lymphopenic environments, promoting the reconstitution of the lymphoid compartment in the periphery (Surh and Sprent, 2008). Interestingly, we showed a higher ratio of cDCs versus CD4⁺ T cell, and, to a lesser extent versus CD8⁺ T cell, in STING GOF mice compared to their WT controls (Fig. 3C), while cDC numbers were not significantly modified in STING GOF versus control mice (Fig. 3D). This enhanced ratio is therefore linked to a strong decrease of T cell numbers characteristic of STING GOF mice (Fig. 3D). In conclusion, T cells in STING GOF mice have an increased probability to be activated via their TCR considering their low numbers and the unmodified number of cDCs compared to T cells of control mice.

Besides antigenic stimulation, IL-7 could also affect T cell fate in STING GOF T cell lymphopenic environment, since IL-7R engagement promotes T cell homeostatic proliferation (Surh and Sprent, 2008). A marked downregulation of the alpha chain of the

receptor (IL-7Rα), reminiscent of IL-7R engagement, was observed for both CD4⁺ and CD8⁺ T cells (Fig. 3E). IL-7Rα expression was significantly downregulated in CD4⁺ and CD8⁺ naïve and memory T cells from STING GOF mice (Fig. EV5A,C), indicating that T cell lymphopenia impacts T cell fate at early stage in the periphery of STING GOF mice. In addition, we observed a statistically significant decrease of *il7ra* (encoding IL7Rα) and *cdkn1b* (encoding the p27ᵏⁱᵖ1 cyclin-dependent kinase inhibitor) transcripts, an increase of *bcl2* transcript (statistically significant for CD8⁺ T cells), and a statistically significant increase of *gfi1* (encoding GFI1, growth-factor independent 1) and *socs1* (encoding SOCS-1 (suppressor of the cytokine signaling 1) transcripts, in the RNA-seq analysis (Appendix Table S2), all of which are also signs of IL-7-R engagement (Surh and Sprent, 2008; Mazzucchelli and Durum, 2007; Winer et al, 2022). Therefore, the results are in favor of an increased IL-7R engagement, and we can suggest that STING GOF T cells are more activated by IL-7 because they will encounter

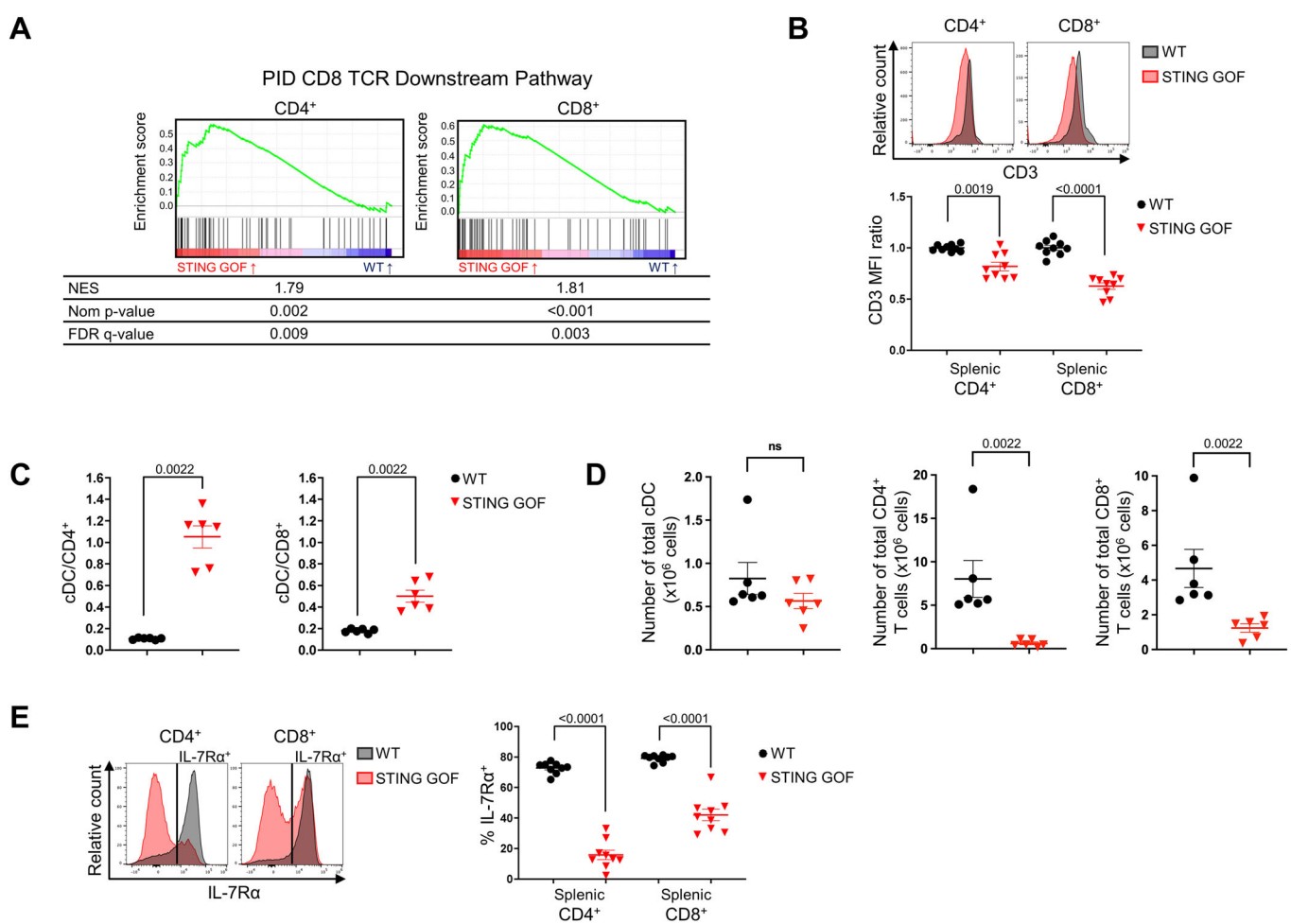

**Figure 3. STING GOF T cells displayed features of TCR and IL-7R engagement mediated by the lymphopenic environment.**

(A) GSEA of the CD8 TCR downstream pathway signature (PID) among genes deregulated in STING GOF versus WT CD4+ or CD8+ T cells. Enrichment plots, normalized enrichment score (NES), nominal p value (Nom p value) and false discovery rate q value (FDR q value) are shown for each analysis. (B) Ratio of CD3 mean fluorescence intensity (MFI) on splenic total CD4+ or CD8+ T cells from STING GOF mice and their WT littermate controls. Ratio was normalized on the mean of WT controls of each analysis. Representative histograms are shown. (C, D) Ratio of cDCs absolute number per CD4+ or CD8+ T cells numbers (C) and corresponding absolute numbers (D) from the spleen of STING GOF mice and their WT littermate controls. (E) Proportion of IL-7Rα-expressing cells among splenic total CD4+ or CD8+ T cells from STING GOF mice and their WT littermate controls. Representative histograms are shown. Each data point corresponds to one mouse; mean ± SEM are shown per population for six to nine mice from two (C, D) and three (B, E) independent experiments (biological replicates). Statistical significances are calculated with a two-tailed Mann–Whitney test; ns (non-significant), P > 0.05.

an abnormal amount of IL-7 per T cell, compared to a physiological situation, as described in lymphopenia-induced proliferation.

Taken together, our results suggest that the lymphopenic environment impacts T cells in STING GOF mice through TCR and IL-7R engagement since the naive T cell stage, pathways which are known to govern lymphopenia-induced homeostatic proliferation.

## T cell exhaustion phenotype of STING GOF T cell is reversed in a non-lymphopenic environment

We next wondered whether STING GOF T cells would display an exhausted profile in a non-lymphopenic context. To this purpose, we took advantage of bone marrow transplantation experiments. We sorted long term-hematopoietic stem cells (LT-HSCs) from bone marrow (BM) of donor STING GOF or WT littermate control mice (CD45.2+, donor) and

co-transplanted them along with supportive WT BM cells (CD45.1+, support) into irradiated WT recipient mice (CD45.1+CD45.2+, host) (Fig. 4A). First, we confirmed the hematopoietic intrinsic effect of the STING GOF mutation on splenic T cell reconstitution (Motwani et al, 2019), as very few T cells were derived from the STING GOF LT-HSCs (red dots), while reconstitution was effective from WT LT-HSCs (black dots). However, the lack of T cell reconstitution from the STING GOF LT-HSCs was compensated by transplantation of supportive WT BM-derived T cells (gray dots) (Fig. 4B). Consequently, absolute numbers of total (including both CD45.1+ and CD45.2+) CD4+ and CD8+ T cells were similar in STING GOF (STING GOF→WT) and WT LT-HSCs (WT→WT) transplantation settings (Fig. 4C). Interestingly, STING GOF LT-HSCs-derived CD4+ and CD8+ T cells did not overexpress the inhibitory receptors PD-1, TIGIT, TIM-3 or LAG-3 when compared to WT LT-HSCs-derived T cells, or to WT supportive BM-derived T cells (Fig. 4D). Consistent with this, there were no terminally exhausted (TE)

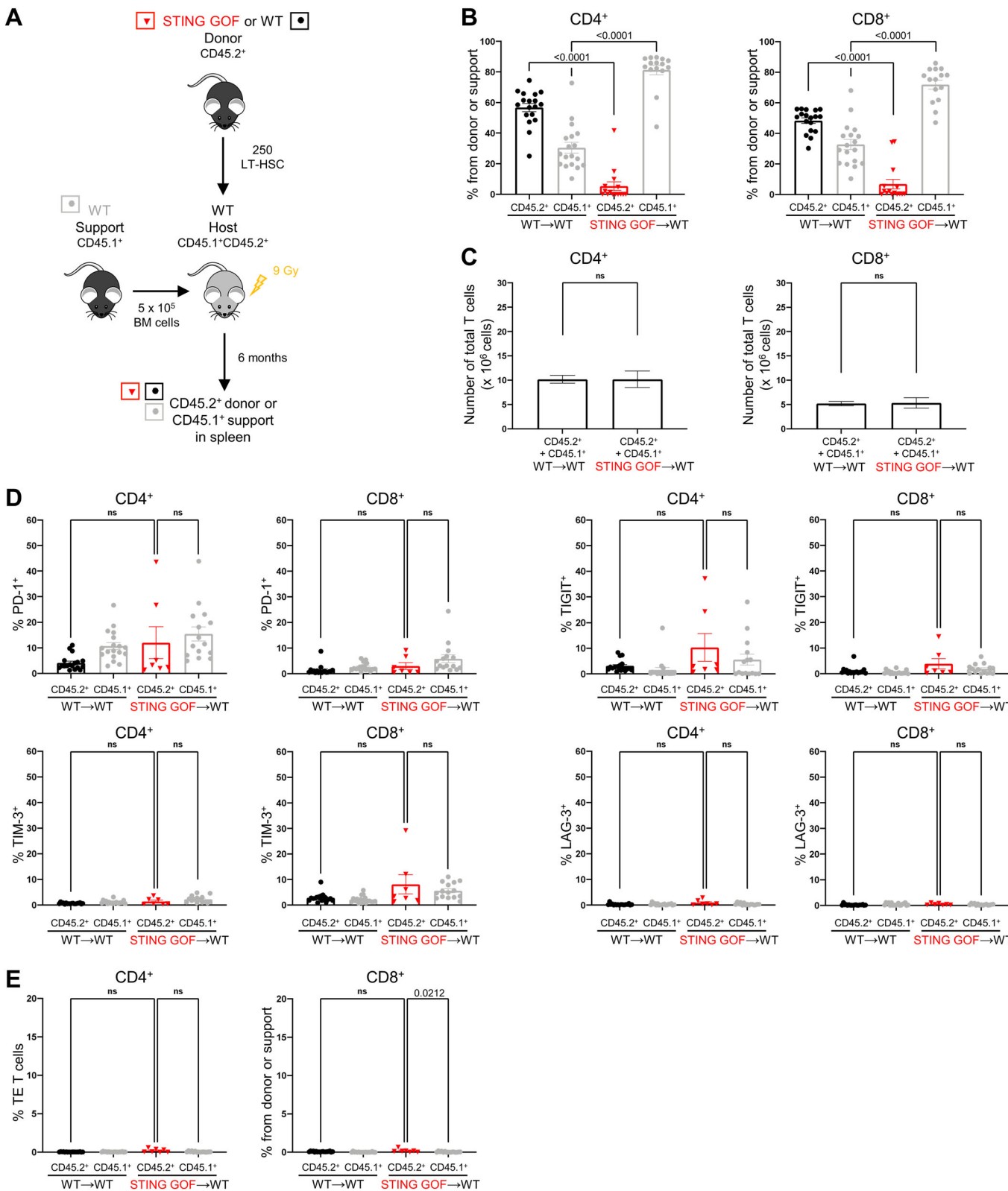

**Figure 4.   T cell exhaustion phenotype of STING GOF T cell is reversed in a non-lymphopenic environment.**

(A–E) LT-HSCs from STING GOF mice and their WT littermate controls (CD45.2⁺) were transplanted with WT BM supportive cells (CD45.1⁺) in WT irradiated hosts to generate STING GOF→WT and WT→WT mice. Six months later, spleens were assessed for T cell reconstitution and immunophenotyping. (**A**) Strategy of LT-HSCs transplantations with supportive BM cells. (**B**) Proportion of cells derived from donor (CD45.2⁺) LT-HSCs or WT BM supportive cells (CD45.1⁺) among splenic CD4⁺ or CD8⁺ T cells from STING GOF→WT and WT→WT mice. (**C**) Absolute numbers of total (CD45.2⁺ and CD45.1⁺) splenic CD4⁺ or CD8⁺ T cells from STING GOF→WT and WT→WT mice. (**D**) Proportion of PD-1-, TIGIT-, TIM-3- and LAG-3-expressing cells among splenic CD4⁺ or CD8⁺ T cells. (**E**) Proportion of terminally exhausted (TE) T cells among splenic CD4⁺ or CD8⁺ T cells. Each data point corresponds to one mouse; mean ± SEM are shown per population for seven to eighteen mice from three independent experiments (biological replicates). Statistical significances are calculated with a two-tailed Mann–Whitney test; ns (non-significant), $P > 0.05$. Source data are available online for this figure.

T cells among STING GOF LT-HSC-derived CD4⁺ and CD8⁺ T cells (Fig. 4E).

Nevertheless, because of the WT background of recipient mice in LT-HSC transplantations, radioresistant stroma was composed of WT cells (Fig. 4A), which is not the case in STING GOF mice. In order to determine the potential impact of the STING GOF stroma on the T cell exhaustion phenotype, we performed transplantation experiments in which total BM cells from WT donor mice (CD45.1⁺, donor) were transplanted into STING GOF or WT irradiated recipient mice (CD45.2⁺, host) (Fig. EV6A). As expected, a good splenic T cell reconstitution was observed upon transplantation of WT donor BM cells into STING GOF irradiated host (Fig. EV6B). In addition, the absolute numbers of CD4⁺ and CD8⁺ T cells were comparable in the two groups (WT→WT, and WT→STING GOF) of mice (Fig. EV6C). Moreover, the proportion of CD4⁺ and CD8⁺ T cells expressing the inhibitory receptors PD-1, TIGIT, TIM-3 or LAG-3 was similar following transplantation of WT donor BM cells into irradiated STING GOF mice or WT recipients (Fig. EV6D), and the proportion of TE T cells was extremely low in either group of mice (Fig. EV6E).

Together, these data indicate that STING GOF stroma does not trigger a T cell exhaustion phenotype and rather identify T cell lymphopenia as the main mechanism driving T cell exhaustion in STING GOF mice.

## Mice carrying Rag1 hypomorphic mutations also display lymphopenia-associated T cell exhaustion

In order to determine if lymphopenia-mediated T cell exhaustion in STING GOF mice is specific to the STING GOF mutation, we investigated whether another lymphopenic mouse model might also show T cell exhaustion. To this purpose, we studied two mouse models carrying hypomorphic *Rag1* mutations (*Rag1*^R972Q/R972Q^ and *Rag1*^R972W/R972W^ mice, thereafter referred to as RAG1 R972Q and RAG1 R972W mice), also presenting T cell lymphopenia of central origin, based on early block in T cell development (Ott De Bruin et al, 2018) as in STING GOF mice. Indeed, RAG1 R972Q and R972W mutant mice had similar levels of CD4⁺ and CD8⁺ T cell lymphopenia in spleen, compared to STING GOF mice (Fig. EV7A).

Interestingly, a significant proportion of CD4⁺ and CD8⁺ T cells from RAG1 R972Q and R972W mice was expressing inhibitory receptors compared to their WT controls, albeit not to the same extent as observed in STING GOF T mice for the majority of inhibitory receptors studied (Fig. 5). TE T cells were also increased among CD4⁺ and CD8⁺ T cells for both RAG1 mutants compared to their WT controls, without reaching the same proportion as in STING GOF mice (Fig. EV7B). Finally, a significant downregulation of CD3 expression,

consistent with TCR stimulation, was also detected on the surface of T cells in these mice, as for STING GOF mice (Fig. EV7C), and was associated with a significant decrease of IL-7-Rα expression (Fig. EV7D). In conclusion, lymphopenia seems to be sufficient to induce T cell exhaustion. However, as T cell exhaustion appeared to be more severe in STING GOF mice compared to RAG1 hypomorphic mutant mice, additional factors beyond central T cell lymphopenia must contribute to the higher degree of T cell exhaustion observed in STING GOF mice.

## Discussion

Our data revealed an unexpected T cell exhaustion phenotype in STING GOF mice. Usually associated with chronic T cell stimulation (as in the case of chronic viral infections and cancers), this altered state of T cell differentiation turned out to be of particular interest since it could account for some of the functional defects of mature STING GOF T cells, especially their lack of proliferation in response to TCR activation (Bouis et al, 2019). Moreover, we show here that T cell exhaustion in STING GOF mice is type I IFN-independent, as was the T cell proliferation defect (Bouis et al, 2019). Overall, T cell exhaustion could contribute to T cell proliferation defects and apoptosis, in addition to intrinsic effects of the STING protein in mature T cells (Cerboni et al, 2017; Gulen et al, 2017; Wu et al, 2019; Concepcion et al, 2022).

T cell exhaustion is also described as being associated with a progressive loss of effector properties in situations of chronic infection and cancer (McLane et al, 2019). In the case of STING GOF situation, exhausted T cells also appear to limit *Ifng* and *Grzm* expression, which are known to be increased by STING activation (Gao et al, 2022; Benoit-Lizon et al, 2022) and to contribute to lung disease mortality (Gao et al, 2022). Therefore, T cell exhaustion could be beneficial in STING GOF mice by limiting tissue immunopathology, similar to its effect in other autoinflammatory and autoimmune diseases (McKinney et al, 2015). In accordance with this, we also showed an increased proportion of exhausted T cells in the lungs of STING GOF mice. However, it remains unclear whether the severity of the pathology is correlated with a weaker T cell exhaustion phenotype, and other factors are certainly involved in the disease's progression.

In addition to T cell exhaustion associated with prolonged antigenic stimulations such as chronic viral infections or tumoral contexts (Blank et al, 2019), we describe here a state of T cell exhaustion whose main driver is represented by a lymphopenic environment. Like the two other contexts, T cell exhaustion in STING GOF mice appeared to be driven mainly by the environment rather than an intrinsic effect in T cells. Indeed, acute treatment of WT splenocytes with DMXAA in vitro had a

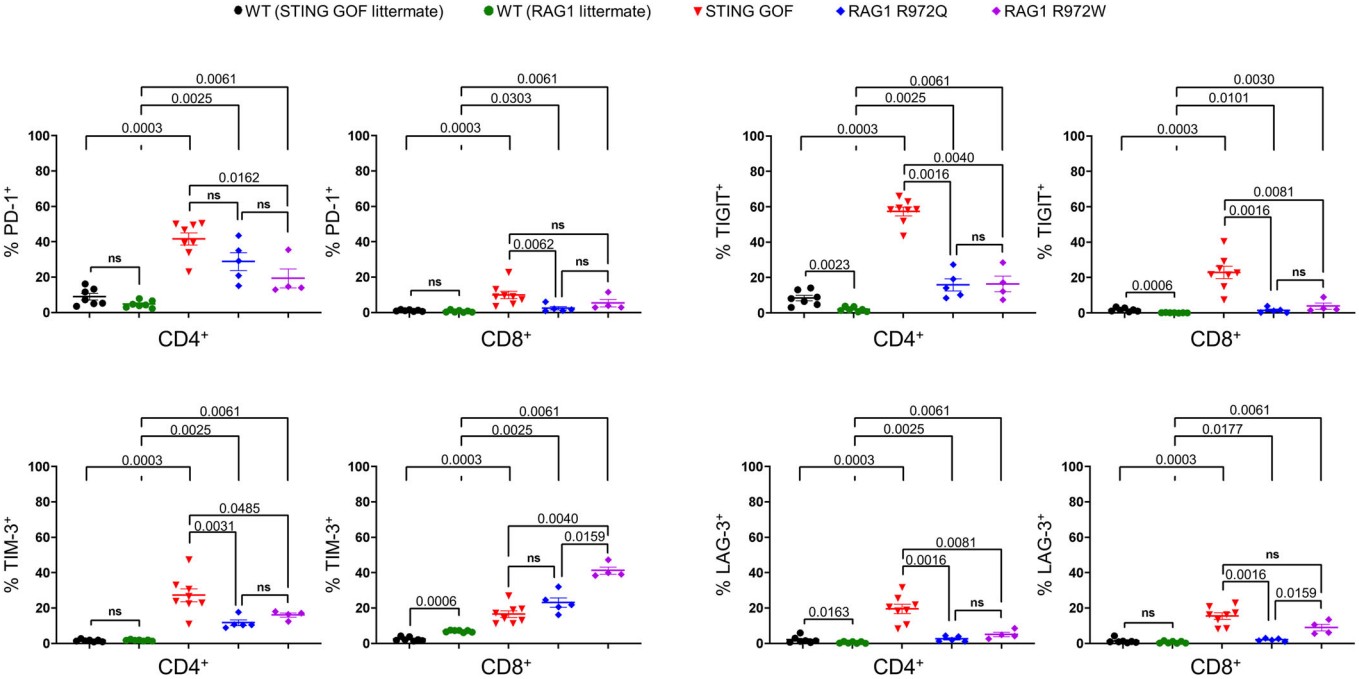

**Figure 5. Mice carrying Rag1 hypomorphic mutations also display lymphopenia-associated T cell exhaustion.**

Proportion of PD-1-, TIGIT-, TIM-3- and LAG-3-expressing cells among splenic CD4+ or CD8+ T cells from hypomorphic RAG1 R972Q and R972W mice and their WT littermate controls by flow cytometry. Data were compared with previous results obtained from STING GOF mice and their WT littermate controls. Each point corresponds to one mouse; mean ± SEM are shown per population for four to eight mice from two independent experiments (biological replicates). Statistical significances are calculated with a two-tailed Mann–Whitney test; ns (non-significant), $P > 0.05$.

pre-activating effect on T cells without inducing an exhaustion phenotype. In addition, bone marrow transplantation experiments showed that STING GOF LT-HSCs-derived T cells developing in a predominantly WT and non-lymphopenic environment do not manifest the exhaustion phenotype. Among environmental factors potentially implicated in the T cell exhaustion of STING GOF mice, type I IFNs were shown to be dispensable, despite the potential role of these cytokines suggested in the literature (Sumida et al, 2022; Terawaki et al, 2011; Wilson et al, 2013). When studying other environmental factors, the STING GOF stroma does not appear to be sufficient to induce T cell exhaustion.

STING GOF T cells display signs of TCR and IL-7R engagement since the naive T cell stage, consistent with lymphopenia-associated signals provided by (auto)antigen and IL-7-mediated stimulation, respectively. While these signals are known to lead to the generation of memory-like T cells through homeostatic proliferation (Surh and Sprent, 2008), T cell exhaustion appeared as a new phenomenon, presumably linked to the persistence of these signals in a context of early and profound lymphopenia. Finally, the observation that mice carrying hypomorphic *Rag1* mutations are lymphopenic and display obvious signs of T cell exhaustion further supports the notion that lymphopenia may represent an important mechanism leading to T cell exhaustion. Altogether, our results suggest a model in which signals engaged in STING GOF T cells (i.e., TCR engagement, as shown by several GSEA signatures and CD3 downregulation, and IL-7-R engagement, as shown by downregulation of IL-7-Rα and RNA-seq data), due to the profound lymphopenic environment, as soon as they leave the thymus and gain the periphery at the naive stage, precedes the setting of the exhaustion

phenotype, the latter being highly reinforced at the memory stage, where all inhibitor receptors, and TIM-3 in particular, are highly expressed. This is consistent with the model of T cell homeostatic proliferation described in lymphopenic contexts.

Nevertheless, T cell exhaustion in STING GOF mice appears to be more severe than the one observed in the two RAG1 hypomorphic mutant models studied. This difference does not depend on the severity of the lymphopenia, since the latter was even more profound, especially in the RAG1 R972W mutant. Thus, the STING GOF mutation may synergize with lymphopenia in promoting and aggravating T cell exhaustion. An important contributory mechanism to the phenomenon may be represented by the intrinsic effect of STING on T cell activation, as suggested by the upregulation of the $Ca^{2+}$-NFAT pathway upon DMXAA treatment. This NFAT activation had been already highlighted by a previous transcriptomic analysis, also indicating its independence from type I IFN (Wu et al, 2020). We assume that STIM1 may play a role in this process, as it has been described as an ER anchor protein for STING (Srikanth et al, 2019). In the case of STING GOF, the STING protein localizes to the ERGIC and no longer binds STIM1. This could, therefore, allow STIM1 to traffic to the ER–plasma membrane junction, facilitating Store-Operated $Ca^{2+}$ Entry (SOCE), leading to $Ca^{2+}$ influx into the cell and activation of the NFAT pathway.

In conclusion, the STING GOF mutation could place T cells in a pre-activated state, making them not only more sensitive to endoplasmic reticulum stress and apoptosis (Wu et al, 2019), but also to exhaustion. Even if we demonstrated that STING GOF radioresistant cells (stroma) do not trigger a T cell exhaustion phenotype by themselves, synergistic effects of the STING GOF

mutation on the antigen presentation capacity of both non-hematopoietic and hematopoietic cells, especially cDC, could reinforce TCR stimulation in the lymphopenic context and thus contribute to exhaustion. The impact of this mutation on major histocompatibility complex (MHC) molecule expression has already been demonstrated in endothelial cells within the lung environment in STING GOF mice (Gao et al, 2022). Moreover, STING activation by agonists in mice has been associated with the loss of cDC1 (Jneid et al, 2023), a subpopulation that plays a crucial role in maintaining and isolating progenitor exhausted (PE) T cells and preventing their transition into terminally exhausted (TE) T cells (Dähling et al, 2022).

Overall, our data show that the T cell exhaustion phenotype in STING V154M mice is the result of poor central production of T cells and will potentially reinforce this lymphopenia by a self-maintaining loop. These data should be considered in the context of severe T cell lymphopenia as severe combined immunodeficiency (SCID). Importantly, recent descriptions of SCID patients developing T cell exhaustion have reinforced a lymphopenia-mediated T cell exhaustion mechanism (Labrosse et al, 2023; Dong et al, 2023). In particular, the analysis of 61 SCID patients receiving hematopoietic cell transplantation (HCT) recently showed that patients with poor T cell reconstitution after HCT display an increased frequency of exhausted T cells, compared to patients with normal T cell counts after HCT, therefore directly linking lymphopenia with T cell exhaustion (Labrosse et al, 2023). In addition, T cell depletion treatments were identified as an important factor associated with PD-1 expression on T cells in patients receiving HCT (Simonetta et al, 2019).

In conclusion, the data and insights gained from the STING GOF mice model could therefore be of key importance for the follow-up of patients receiving HCT, immunosuppression, or CAR-T cell therapy for T cell malignancies, as well as for a better comprehension of potential T cell exhaustion in lymphopenic murine models. Finally, STING GOF mice constitute a good model for the study of T cell exhaustion in non-infectious and non-tumoral contexts.

# Methods

### Reagents and tools table

| Reagent/resource | Reference or source | Identifier or catalog number |
|---|---|---|
| **Experimental models** | | |
| STING$^{V154M/+}$ (*M. musculus*) | Bouis, et al, 2019 | C57BL/6-Sting$^{em1Psou}$/Orl |
| IFNAR1 knockout (*M. musculus*) | Müller, et al, 1994 | B6(Cg)-*Ifnar1tm1.2Ees*/J |
| RAG1$^{R972Q/R972Q}$ (*M. musculus*) | Ott De Bruin, et al, 2016 Ott De Bruin, et al, 2018 | |
| RAG1$^{R972W/R972W}$ (*M. musculus*) | Ott De Bruin, et al, 2016 Ott De Bruin, et al, 2018 | |
| **Recombinant DNA** | | |
| N/A | | |
| **Antibodies** | | |

| Reagent/resource | Reference or source | Identifier or catalog number |
|---|---|---|
| **Antibodies used for flow cytometry** | | |
| FITC anti-mouse CD3e, 1:400 | BD Pharmingen | 553062 |
| PerCP Cy5.5-anti-mouse CD3e, 1:100 | BD Pharmingen | 551163 |
| PE-Cy7 anti-mouse CD3e, 1:100 | BD Pharmingen | 552774 |
| AF700 anti-mouse CD4, 1:1000 | BD Pharmingen | 557956 |
| FITC anti-mouse CD8a, 1:400 | BD Pharmingen | 561966 |
| PE anti-mouse CD8a, 1:400 | BD Pharmingen | 553032 |
| PE-CF594 anti-mouse CD8a, 1:200 | BD Pharmingen | 562283 |
| APC anti-mouse CD44, 1:1000 | BD Pharmingen | 559250 |
| AF594 anti-mouse CD44, 1:300 | BioLegend | 103054 |
| APC Cy7 anti-mouse CD45.1, 1:100 | Invitrogen | A15415 |
| PE-Cy7 anti-mouse CD45.2, 1:100 | eBioscience | 25-0454-82 |
| FITC anti-mouse CD48, 1:100 | BioLegend | 103403 |
| PE anti-mouse CD62L, 1:600 | BD Pharmingen | 553151 |
| PE-Cy7 anti-mouse CD62L, 1:100 | BioLegend | 104418 |
| APC anti-mouse CD117, 1:400 | BD Pharmingen | 553356 |
| PE-Dazzle594 anti-mouse CD127, 1:100 | BioLegend | 135032 |
| PE anti-mouse CD150 (SLAM), 1:100 | BioLegend | 162606 |
| APC anti-mouse CD223, 1:100 | BD Pharmingen | 562346 |
| PE anti-mouse CD279, 1:200 | BD Pharmingen | 561788 |
| PE-CF594 anti-mouse CD279, 1:100 | BD Pharmingen | 562523 |
| PE anti-mouse CD366, 1:200 | BD Pharmingen | 119703 |
| PE-Cy7 anti-mouse Sca-1 (Ly-6A/E), 1:400 | BioLegend | 108114 |
| APC anti-mouse TIGIT, 1:400 | BioLegend | 142105 |
| AF647 anti-mouse TCF-1, 1:200 | BD Pharmingen | 566693 |
| PE anti-mouse NFATc1, 1:100 | BioLegend | 649606 |
| **Antibodies used for LT-HSC sorting** | | |
| Biotin anti-mouse CD3, 1:50 | BioLegend | 100243 |
| Biotin anti-mouse CD4, 1:1000 | BioLegend | 100403 |
| Biotin anti-mouse CD8, 1:100 | eBioscience | 13-0081-82 |
| Biotin anti-mouse CD11b, 1:8000 | BioLegend | 101204 |
| Biotin anti-mouse CD19, 1:800 | BioLegend | 152420 |
| Biotin anti-mouse CD45R/ B220, 1:1000 | BioLegend | 103203 |
| Biotin anti-mouse CD49b, 1:200 | BioLegend | 103521 |

| Reagent/resource | Reference or source | Identifier or catalog number |
|---|---|---|
| Biotinylated anti-mouse Ly-6G/C, 1:200 | BioLegend | 108403 |
| Biotinylated anti-TER-119 (TER-119), 1:200 | BioLegend | 116203 |
| **Oligonucleotides and other sequence-based reagents** | | |
| N/A | | |
| **Chemicals, enzymes and other reagents** | | |
| RPMI-1640 with L-glutamine | Lonza | 11635150 |
| PBS | Gibco | 11503387 |
| Fetal bovine serum | Invitrogen | A5209402 |
| Foxp3/Transcription factor staining buffer set | Thermo Fisher Scientific | 11500597 |
| Fixable Viability Dye eFluor™ 450 | Thermo Fisher Scientific | 65-0863-14 |
| DAPI | Sigma-Aldrich | 10236276001 |
| RNeasy® Plus Micro Kit | Qiagen | 74034 |
| SMARTer Stranded Total RNA-Seq Kit, Pico Input Mammalian | Takara | 634411 |
| NextSeq 500/550 High Output Kit V.2 | Illumina | FC-420-1001 |
| Fura Red™ acetoxymethyl ester (AM) | Thermo Fisher Scientific | F1221 |
| Pluronic F-127 | Thermo Fisher Scientific | P3000MP |
| Ionomycin | InvivoGen | inh-ion |
| β-mercaptoethanol | Gibco | 11528926 |
| penicillin/streptomycin | Gibco | 15140122 |
| HEPES | Lonza | CC-5024 |
| 5,6-dimethylxanthenone-4-acetic acid (DMXAA) | InvivoGen | tlrl-dmx |
| Streptavidin AF700 | Invitrogen | S21383 |
| DNase I (DN25) | Sigma-Aldrich | 9003-98-9 |
| Collagenase (Liberase) | Sigma-Aldrich | 5401054001 |
| NucleoSpin RNA Plus XS kit | Macherey–Nagel | 740990.50 |
| Maxima H Minus cDNA Synthesis Master Mix | Thermo Fisher Scientific | M1662 |
| TaqMan™ PreAmp Master Mix | Thermo Fisher Scientific | 4488593 |
| TaqMan™ expression Master Mix | Thermo Fisher Scientific | 4369016 |
| **TaqMan gene expression assays used in qPCR** | | |
| TaqMan gene expression assay for *Ifng* | Thermo Fisher Scientific | Mm00434256_m1 4453320 |
| TaqMan gene expression assay for *Gzmb* | Thermo Fisher Scientific | Mm01168134_m1 4453320 |
| TaqMan gene expression assay for *GAPDH* | Thermo Fisher Scientific | Mm99999915_g1 4453320 |
| TaqMan gene expression assay for *Actin beta* | Thermo Fisher Scientific | Mm00607939_s1 4453320 |
| **Software** | | |

| Reagent/resource | Reference or source | Identifier or catalog number |
|---|---|---|
| FlowJo (TreeStar) | https://www.flowjo.com | |
| GraphPad Prism 8.2.1 | https://graphpad.com | |
| GSEA (UC San Diego, Broad Institute) | https://www.gsea-msigdb.org/gsea/index.jsp | |
| **Other** | | |
| Attune NxT™ flow cytometer | Thermo Fisher Scientific | |
| FACSAria™ Fusion flow cytometer | BD | |
| Illumina NextSeq 500 | Illumina | |
| NanoPhotometer™ N60 | Implen | |
| CFX Connect Real-Time PCR Detection System | Bio-Rad | |

## Mice and cells

The STING$^{V154M/+}$ mice (called thereafter STING GOF mice) were generated as previously described (Bouis et al, 2019). Mice were bred and maintain in specific opportunistic pathogen-free (SOPF) conditions at the animal facility of the Molecular and Cellular Biology Institute (IBMC, Strasbourg, France). Littermate wild-type (WT) and sex- and age-matched mice were always used as control animals. STING GOF mice were crossed with IFNAR1 (IFN-α/β receptor chain 1) knockout (KO) mice, thereafter named IFNAR KO mice (provided by TAAM, CNRS, Orleans, France) (Müller et al, 1994). All animal experiments were performed with the approval of the "Direction départementale des services vétérinaires" (Strasbourg, France), and protocols were approved by the ethics committee ("Comité Régional d'Éthique en Matière d'Expérimentation Animale de Strasbourg", CREMEAS) under relevant institutional authorization ("ministère de l'Éducation Nationale, de l'Enseignement Supérieur et de la Recherche"; authorization APAFIS #2387-2015072907553237). The RAG1$^{R972Q/R972Q}$ and RAG1$^{R972W/R972W}$ mice (called thereafter RAG1 hypomorphic mice) have been previously described (Ott De Bruin et al, 2018, 2016), and experiments with these mice were performed according to protocol ASP LCIM6E, approved by the National Institute of Allergy and Infectious Diseases' animal care and use committee. The genetic background of all mice used in this study is C57BL/6. Mice were analyzed at an age between 2 and 6 months, except for the analysis at 2-week-old (Fig. 2C, see results section and legend), and for LT-HSCs and BM transplantations (see the corresponding Methods paragraphs).

Mice were euthanized by cervical dislocation. Spleen, thymus, and bone marrow (BM) were mechanically dilacerated. For pulmonary lymphoid infiltrates analysis, lungs were cut into small pieces and digested with an enzyme solution containing DNase and collagenase for 45 min at 37 °C. Red blood cells were removed by osmotic lysis using ACK (Ammonium-Chloride-Potassium) buffer.

## Flow cytometry and cell sorting

Splenic, thymic, BM or lung single-cell suspensions were analyzed by flow cytometry according to standard protocol. Cells were stained with the antibodies listed in the Reagents and Tools table in

PBS (Gibco) supplemented with 2% v/v FBS (Dutscher). For intracellular staining of total (nuclear and cytosolic) TCF-1 AF647 and NFATc1 PE, cells were fixed and permeabilized using the Foxp3/Transcription Factor Staining Buffer Set (Thermo Fisher Scientific), according to the manufacturer's instructions. Cell viability was assessed using Fixable Viability Dye eFluor™ 450 (eBioscience) or DAPI (1 µM/mL Sigma) stainings.

Flow cytometric acquisition was performed on an Attune NxT™ flow cytometer (Thermo Fisher Scientific), and data were analyzed using FlowJo software (TreeStar). Cell sorting was carried out on a BD FACSAria™ Fusion flow cytometer (IGBMC Flow Cytometry Facility, Strasbourg, France). Sorted populations displayed a purity ≥95%.

## RNA-seq on sorted CD4+ and CD8+ T cells

Splenocytes from STING GOF mice ($n = 5$) and WT littermate controls ($n = 5$) were stained as described above with anti-CD3e FITC, anti-CD4 AF700, anti-CD8a PE and DAPI (1 µM/mL, Sigma) before positive sorting of live CD3+CD4+ or CD3+CD8+ T cells by flow cytometry. Total RNA was extracted using RNeasy® Plus Micro Kit (Qiagen) according to the manufacturer's instructions. RNA-seq analyses were performed by the Genomax Facility (INSERM U1109, ImmunoRhumatologie Moléculaire, Université de Strasbourg, France). Libraries were prepared from 10 ng RNA using SMARTer Stranded Total RNA-Seq Kit, Pico Input Mammalian (Takara) following the manufacturer's instructions. Briefly, random primers were used for first-strand synthesis, and ribosomal cDNA was cleaved by ZapR V.2 in the presence of mammalian R-probes V.2. Libraries were pooled and sequenced (paired-end $2 \times 75$ bp) on a NextSeq 500 using the NextSeq 500/550 High Output Kit V.2 according to the manufacturer's instructions (Illumina). For each sample, quality control was carried out and assessed with the next-generation sequencing (NGS) Core Tools FastQC. Reads were aligned against the *Mus musculus* mm10 reference genome using TopHat 2 Aligner (Kim et al, 2013) and gene expression levels were estimated using Cufflinks V.2.1.1 (Trapnell et al, 2010). Differential expression analysis was performed with Cuffdiff V.2.1.1 after exclusion of fragments per kilobase million below 9. Gene set enrichment analysis (GSEA) was carried out according to the developer's recommendations. The STING GOF versus WT FPKMs of all genes detected in the RNA-seq were used for the ranked genes list and compared with the indicated signatures from the molecular signatures database (MSigDB), using GSEA Software (UC San Diego, Broad Institute) (Mootha et al, 2003; Subramanian et al, 2005). The statistical significance (nominal P value) of the enrichment score (ES) is estimated by GSEA by using an empirical phenotype-based permutation test procedure that preserves the complex correlation structure of the gene expression data (Subramanian et al, 2005).

## Cytosolic Ca²⁺ level measurements

For cytosolic $Ca^{2+}$ level measurements, splenic single-cell suspensions were stained as described above with Fixable Viability Dye eFluor™ 450 (eBioscience), anti-CD4 AF700 and anti-CD8a PE-CD584 before being loaded with 2 µM Fura Red™ acetoxymethyl ester (AM) (Thermo Fisher Scientific) in PBS 2% FBS containing 0.02% pluronic F-127 (Thermo Fisher Scientific) for 30 min at 37 °C, avoiding light. Loaded cells were washed, resuspended in

DMEM (Gibco) containing 1.8 mM $Ca^{2+}$ and incubated for 10 min at 37 °C, avoiding light, before acquisition using an Attune NxT™ flow cytometer (Thermo Fisher Scientific). After an initial 50-s acquisition sequence, ionomycin (1 µg/mL, InvivoGen) was added, as a positive control, before the acquisition of a second 60-s sequence. $Ca^{2+}$ levels were determined according to time by ratiometric analyzes between VL4 (405 nm-excitation; 660-nm-emission) and BL3 (488-nm-excitation; 695-nm-emission) channels using FlowJo software (TreeStar).

## Splenocyte culture

In vitro splenocytes stimulations were performed in complete RPMI-1640 medium containing L-glutamine (Lonza) supplemented with 10% v/v FBS (Dutscher), 50 mmol/L β-mercaptoethanol (Gibco), 1% penicillin/streptomycin (Gibco), and 10 mmol/L HEPES (Lonza). Splenocytes were stimulated for the indicated times at 37 °C, under 5% $CO_2$, with the STING agonist 5,6-dimethylxanthenone-4-acetic acid (DMXAA) (10 µg/mL, InvivoGen).

## LT-HSCs transplantations with supportive BM cells

For LT-HSCs sorting, total BM cells were stained with a lineage cocktail of biotinylated antibodies (listed in the Reagents and Tools table), followed by depletion using Dynabeads sheep anti-rat IgG beads (Thermo Fisher Scientific). Lineage-negative cells were stained as described above with Streptavidin AF700 (Invitrogen), anti-CD48 FITC, anti-CD117 (c-Kit) APC, anti-CD150 (SLAM) PE, anti-Sca-1 (Ly-6A/E) PE-Cy7 and DAPI (1 µM/mL, Sigma) before positive sorting of live LT-HSCs (CD48⁻CD150+Lin⁻Sca1⁻cKit+) by flow cytometry. Isolated LT-HSCs were at least 98% pure. 250 STING GOF or WT LT-HSCs from 6-week-old CD45.2+ donor mice were i.v. injected with $5 \times 10^5$ supporting CD45.1+ WT BM cells into 9-Gy lethally irradiated CD45.1+CD45.2+ WT hosts. The spleens of the mice were harvested 6 months after transplantation, and the splenocytes analyzed by flow cytometry as described above.

## Wild-type (WT) total bone marrow cell transplantations

About $1 \times 10^6$ WT total BM cells from 10 to 15-week-old CD45.1+ donor mice were i.v. injected into 9-Gy lethally irradiated CD45.2+ STING GOF or WT hosts. Mouse spleens were harvested 6 months after transplantation and the splenocytes analyzed by flow cytometry as described above.

## Quantitative PCR on sorted exhausted and non-exhausted CD4+CD8+ T cells

Exhausted (TIM-3+PD-1+) and non-exhausted (TIM-3⁻PD-1⁻) CD4+CD8+ T cells were sorted from mouse spleens by flow cytometry for downstream gene expression analysis. Total RNA was extracted using the NucleoSpin RNA Plus XS kit (Macherey–Nagel) and quantified with a NanoPhotometer™ N60 (Implen). Reverse transcription was performed using the Maxima H Minus cDNA Synthesis Master Mix (Thermo Fisher Scientific). The resulting cDNA was pre-amplified with the TaqMan™ PreAmp Master Mix (Thermo Fisher Scientific), using a pooled mix of

TaqMan Gene Expression Assays (listed in the Reagents and Tools Table). Quantitative PCR was then performed using the same individual TaqMan Gene Expression Assays and gene TaqMan™ expression Master Mix (Thermo Fisher Scientific) on a CFX Connect Real-Time PCR Detection System (Bio-Rad). Gene expression levels were normalized to housekeeping genes using the ΔΔCt method (listed in the Reagents and Tools Table).

## Statistical analysis

No animal, sample, or data were excluded from the analysis. No blinding was performed. All data were presented as means ± SEMs. Statistical analyses were performed with Prism 8.2.1 software (GraphPad). Taking into account that the sample size was ≤30 and didn't follow a normal distribution, statistical significance was calculated with the non-parametric test of Mann–Whitney test, except for Figs. 1H and 2B, where statistical significance was calculated with the Wilcoxon signed-rank test with a hypothetical value of 1 (see details in each Figure legends), and except for GSEA analysis (Figs. 1A, 3A and EV4A) (see methods section for RNA-seq). *P* values are directly indicated in the graphs in each figure. Non-significant comparisons are labeled "ns".

## Adherence to community standards

ARRIVE checklist (https://arriveguidelines.org) has been followed to conduct the study and write the manuscript.

---

### The paper explained

#### Problem
Gain-of-function (GOF) mutations in the STING gene cause STING-associated vasculopathy with onset in infancy (SAVI), a severe auto-inflammatory disease. In a mouse model carrying one of these mutations, a major reduction in T cells is observed, partly due to defective T cell development in the thymus. However, the mechanisms leading to peripheral T cell dysfunction need to be better described.

#### Results
We studied mature T cells from the spleen of these mutant mice and discovered that they rapidly become "exhausted"—a dysfunctional state with low functionality. This phenotype emerges early in life, but only after peripheral egress of T cells, and does not result from intrinsic STING activation in T cells or stromal cells. Crucially, transplantation of STING GOF hematopoietic stem cells with wild-type bone marrow, which resulted in a non-lymphopenic environment, prevented T cell exhaustion of STING GOF T cells, underscoring lymphopenia as the key driver. In accordance with these results, an exhausted state of T cells in another lymphopenic murine model was also observed.

#### Impact
These findings establish lymphopenia as a central driver of T cell exhaustion, therefore having clinical impact for patients with lympho-penia in different clinical situations.

---

## Data availability

Raw RNA-seq data have been deposited in the EMBL-EBI ArrayExpress archive (accession no. E-MTAB-13658; https://www.ebi.ac.uk/biostudies/ArrayExpress/studies/E-MTAB-13658?query=%20E-MTAB-13658). Source Data for Figs. 1, 3, 5 have been deposited on BioStudies: https://www.ebi.ac.uk/biostudies/studies/S-BSST2071?key=bf3ffafd-8eef-45e9-bfde-15a557d8e3d0.

The source data of this paper are collected in the following database record: biostudies:S-SCDT-10_1038-S44321-025-00292-6.

## Peer review information

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

## Acknowledgements

We thank Fabrice Augé, Christian Galmiche, Delphine Lamon, Manon Lecointe, Fabien Lhericel, Sophie Reibel-Foisset, and Eva Simon for excellent animal care. We thank Vincent Gies, Frédéric Gros, Aurélien Guffroy, Sophie Jung and Thierry Martin (INSERM UMR - S1109, Strasbourg University, France), Vincent Flacher and Christopher Mueller (CNRS UPR 3572, Institute of Molecular and Cellular Biology (IBMC), Strasbourg, France) for scientific discussions. We thank the Flow Cytometry facility of IGBMC (CytoEast) (Illkirch, France) for cell sorting and the Genomax (Strasbourg, France) core facility for RNA-seq analysis. This work was supported by the Hôpitaux Universitaires de Strasbourg (HUS), by the "Direction de la Recherche Clinique et de l'Innovation" (HUS) and by grants from Strasbourg University, by the Agence Nationale de la Recherche (ANR-14-CE14-0026-04, Lumugene; ANR-19-CE15-0028, LYMPHO-STING; and ANR-11-EQPX-022). The study was also supported by the Institut National de la Santé et de la Recherche Médicale (INSERM) and by government grants managed by the Agence Nationale de la Recherche as part of the "Investment for the Future" program (Institut Hospitalo-Universitaire Imagine, grant ANR-10-IAHU-01, Recherche Hospitalo-Universitaire, grant ANR-18-RHUS-0010). Luigi D. Notarangelo is supported by the Division of Intramural Research, National Institute of Allergy and Infectious Diseases, National Institutes of Health.

## Author contributions

**Damien Freytag**: Conceptualization; Data curation; Formal analysis; Validation; Investigation; Visualization; Methodology; Writing—original draft; Writing—review and editing. **Stéphane Giorgiutti**: Conceptualization; Data curation; Formal analysis; Validation; Investigation; Visualization; Methodology; Writing—original draft; Writing—review and editing. **Grégoire Hopsomer**: Conceptualization; Data curation; Formal analysis; Validation; Investigation; Visualization; Methodology; Writing—original draft; Writing—review and editing. **Nadége Wadier**: Resources; Investigation; Methodology. **Sabine Depauw**: Resources; Investigation; Methodology. **Philippe Mertz**: Validation; Investigation; Methodology; Writing—original draft. **Fabrice Augé**: Resources. **Raphaël Carapito**: Resources; Data curation; Investigation; Methodology. **Isabelle Couillin**: Validation; Writing—original draft. **Anne-Sophie Korganow**: Conceptualization; Writing—original draft. **Francesca Pala**: Resources; Investigation; Methodology. **Marita Bosticardo**: Resources; Investigation; Methodology. **Luigi Notarangelo**: Conceptualization; Resources; Writing—original draft. **Frédéric Rieux-Laucat**: Conceptualization; Writing—original draft. **Nicolas Riteau**: Conceptualization; Data curation; Formal analysis; Validation; Investigation; Methodology; Writing—original draft. **Peggy Kirstetter**: Conceptualization; Resources; Data curation; Formal analysis; Validation; Investigation; Visualization; Methodology; Writing—original draft. **Pauline Soulas-Sprauel**: Conceptualization; Data curation; Formal analysis; Supervision; Funding acquisition; Validation; Visualization; Methodology; Writing—original draft; Project administration; Writing—review and editing.

Source data underlying figure panels in this paper may have individual authorship assigned. Where available, figure panel/source data authorship is listed in the following database record: biostudies:S-SCDT-10_1038-S44321-025-00292-6.

## Disclosure and competing interests statement

The authors declare no competing interests.

# Expanded View Figures

**Figure EV1.  T cell exhaustion phenotype in STING GOF mice is acquired during the transition from naive to memory T cells.**

(A, B) Immunophenotyping of splenic T cells from STING GOF mice and their WT littermate controls by flow cytometry. Proportion of PD-1-, TIGIT-, TIM-3- and LAG-3-expressing cells among (A) total $CD4^+$ or $CD8^+$ T cells and (B) naive ($CD44^{low}CD62L^+$), central memory (Cen mem, $CD44^{high}CD62L^+$) and effector memory (Eff mem, $CD44^+CD62L^-$) among $CD4^+$ or $CD8^+$ T cells, from STING GOF mice and their WT littermate controls. Each data point corresponds to one mouse; mean ± SEM are shown per population for four mice from two independent experiments (biological replicates). Statistical significances are calculated with a two-tailed Mann–Whitney test; ns (non-significant), $P > 0.05$.

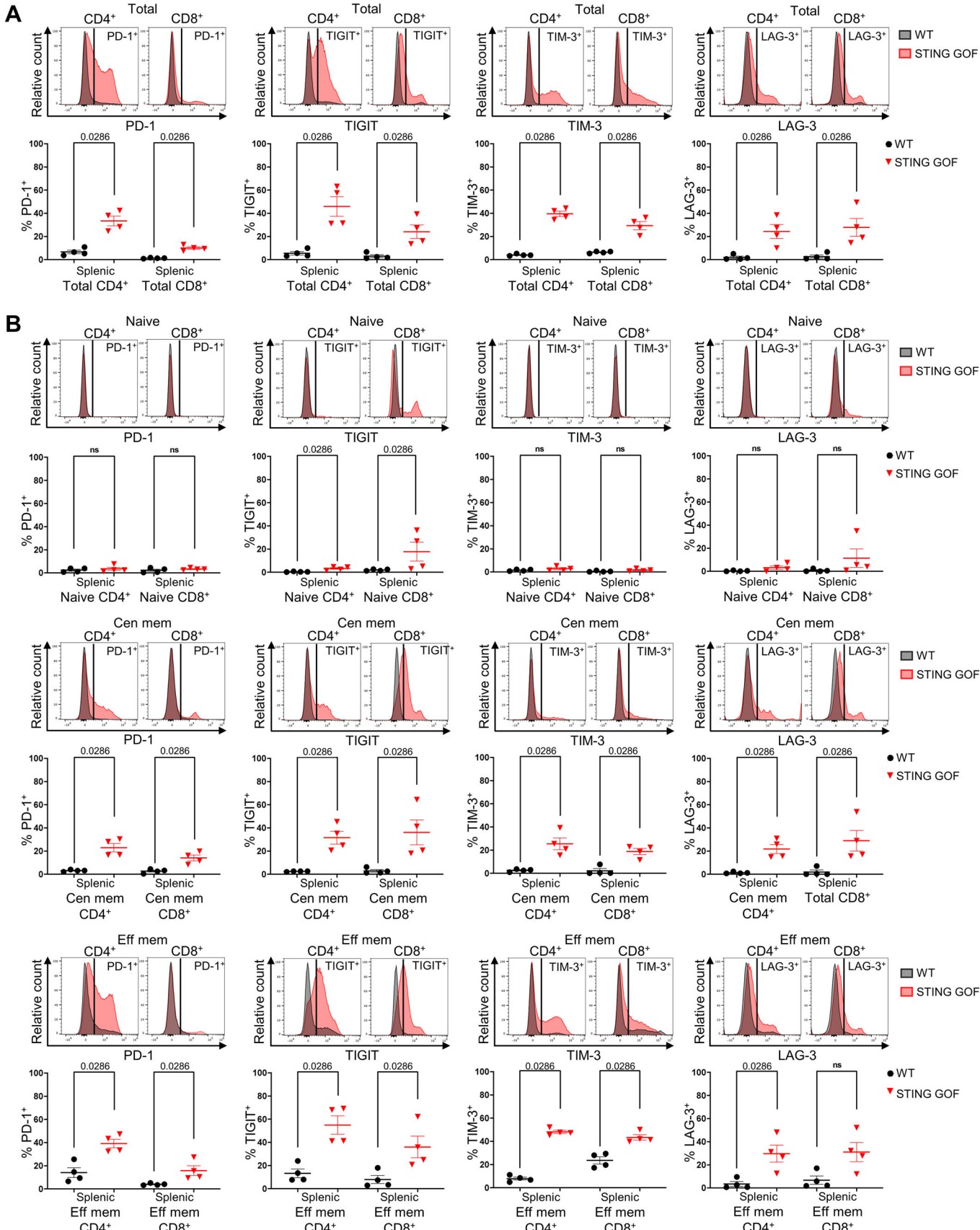

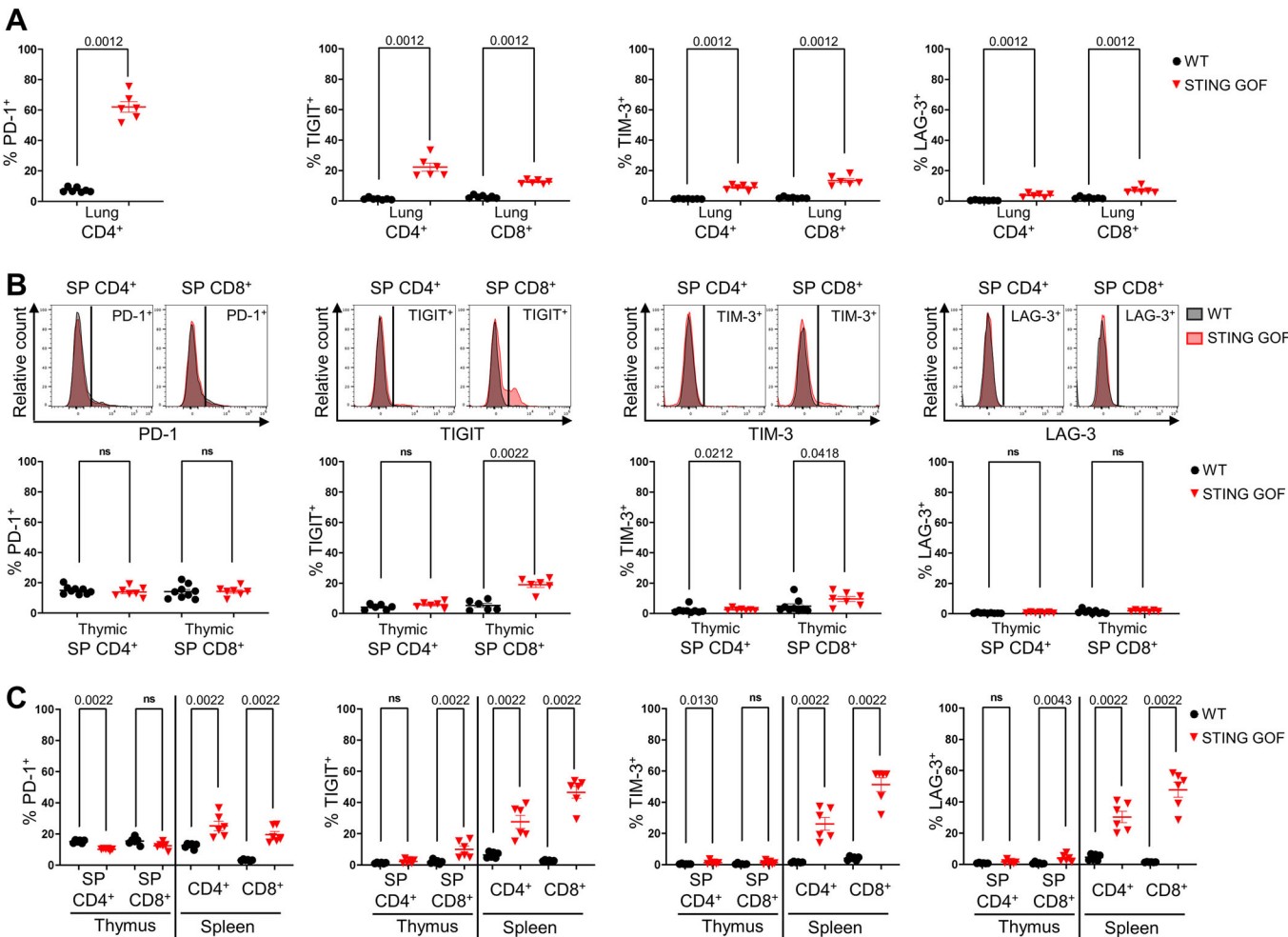

**Figure EV2.  T cell exhaustion in STING GOF mice is acquired early in life and in the peripheral environment.**

(A) Proportion of PD-1-, TIGIT-, TIM-3-, and LAG-3-expressing cells among lung CD4+ or CD8+ T cells from STING GOF mice and their WT littermate controls. (B) Proportion of PD-1-, TIGIT-, TIM-3-, and LAG-3-expressing cells among thymic CD4+ or CD8+ SP T cells from STING GOF mice and their WT littermate controls. Representative histograms are shown. (C) Proportion of PD-1-, TIGIT-, TIM-3-, and LAG-3-expressing cells among thymic and splenic CD4+ or CD8+ SP T cells from 2-week-old STING GOF mice and their WT littermate controls. Each data point corresponds to one mouse; mean ± SEM are shown per population for six to nine mice from two (A) and three (B, C) independent experiments (biological replicates). Statistical significances are calculated with a two-tailed Mann–Whitney test; ns (non-significant), *P* > 0.05.

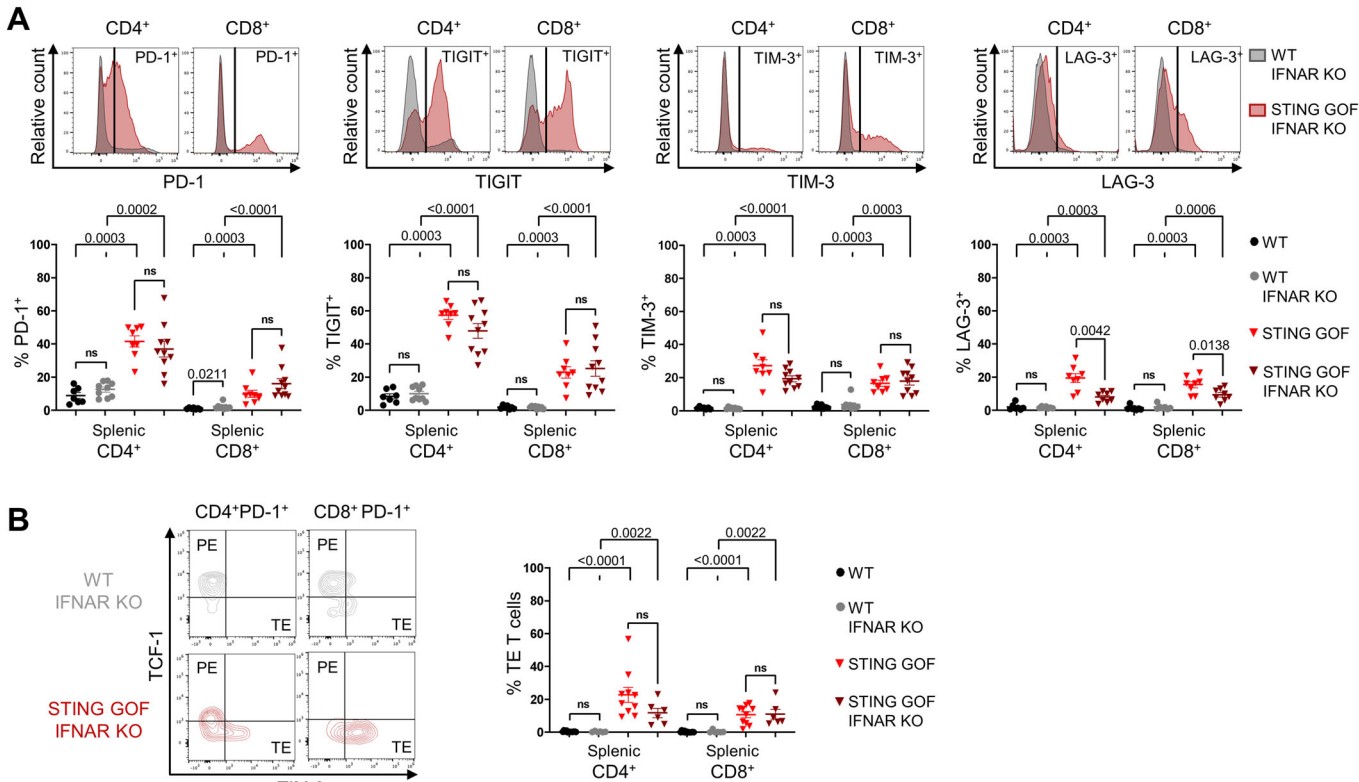

**Figure EV3. T cell exhaustion in STING GOF mice is independent of type I IFNs.**

(A) Proportion of PD-1-, TIGIT-, TIM-3-, and LAG-3-expressing cells among splenic CD4+ or CD8 + T cells from STING GOF IFNAR KO mice and their respective WT littermate controls. Data were compared with the previous result obtained from STING GOF mice and their WT littermate controls (Fig. 1). Representative histograms are shown for the IFNAR KO pair. (B) Proportion of terminally exhausted (TE) T cells among splenic CD4+ or CD8+ T cells from STING GOF IFNAR KO mice and their respective WT littermate controls. Data were compared with results obtained from STING GOF mice and their WT littermate controls (Fig. 1). Representative contour plots are shown for the IFNAR KO pair. Each data point corresponds to one mouse; mean ± SEM are shown per population for six to ten mice from at least three independent experiments (biological replicates). Statistical significances are calculated between IFNAR KO groups as well as between STING GOF groups with a two-tailed Mann–Whitney test.

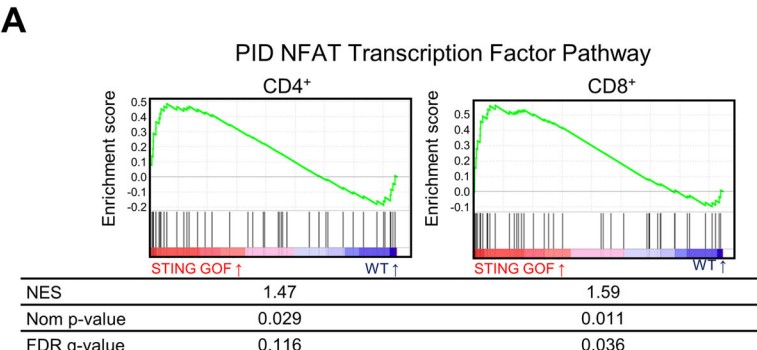

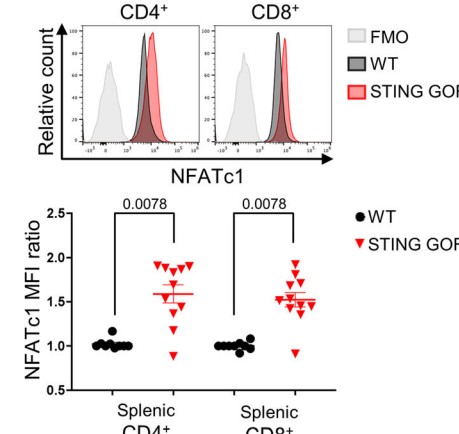

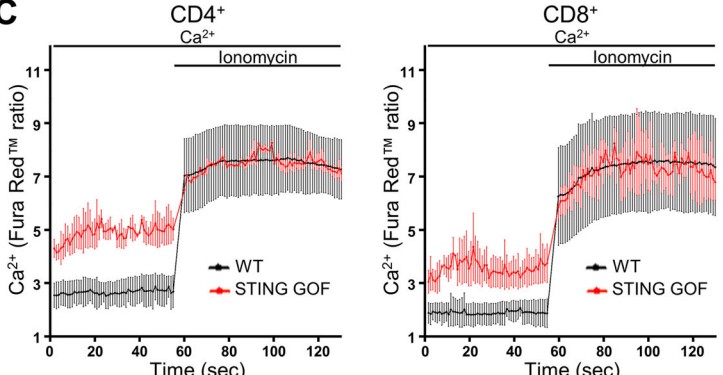

**Figure EV4.  Ca²⁺-NFAT activation in STING GOF T cells.**

(A) GSEA of the NFAT pathway signature (PID) among genes deregulated in STING GOF versus WT CD4⁺ or CD8⁺ T cells. Enrichment plots, normalized enrichment score (NES) and nominal *p* value (*p* value) are shown for each analysis. (B, C) Immunophenotyping and Ca²⁺ levels of splenic T cells from STING GOF mice and their WT littermate controls by flow cytometry. (B) Ratio of total NFATc1 mean fluorescence intensity (MFI) in splenic CD4⁺ or CD8⁺ T cells from STING GOF mice and their WT littermate controls. Ratio was normalized on the WT control of each analysis. Representative histograms are shown. Each data point corresponds to one mouse; mean ± SEM are shown per population for nine to eleven mice from seven independent experiments (biological replicates). Statistical significances are calculated with the Wilcoxon signed-rank test with a hypothetical value of 1: **$P < 0.01$. (C) Relative cytosolic Ca²⁺ levels monitored by Fura Red™ ratio in splenic CD4⁺ or CD8⁺ T cells from STING GOF mice and their WT littermate controls. Splenocytes were recorded in DMEM containing 1.8 mM Ca²⁺ and stimulated with 1 µg/mL ionomycin after 50 s as a positive control. Fura Red™ intensities in VL4 (405 nm-excitation; 660 nm-emission) and BL3 (488 nm-excitation; 695 nm-emission) channels were recorded, corresponding to Ca²⁺-bound and Ca²⁺-free Fura Red™, respectively. Ratio (VL4/BL3) of these Fura Red™ MFI according to time are plotted in a curve graph where each data point represents the mean of two independent experiments (biological replicates), and their error bars the SEM.

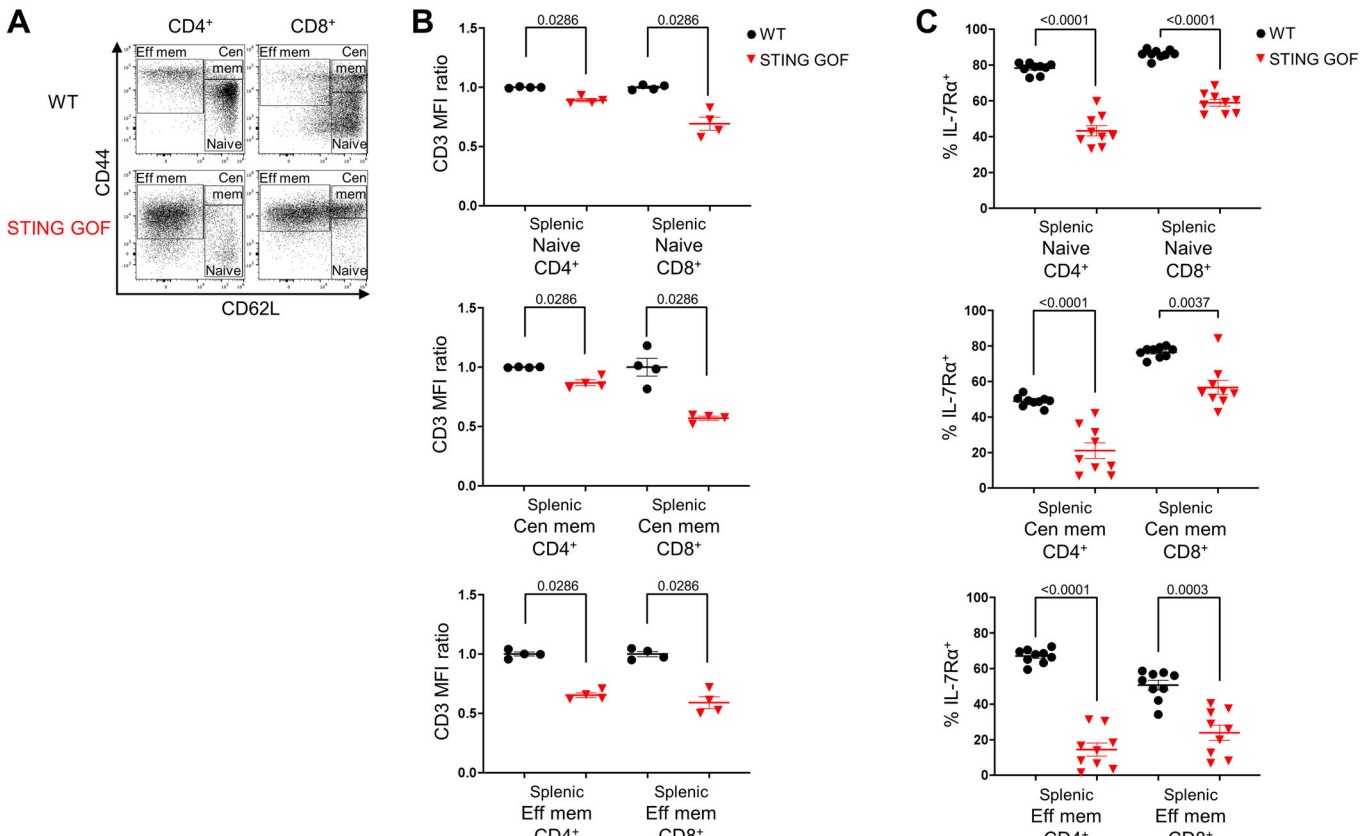

**Figure EV5. TCR and IL-7R engagement since the naive T cell stage in STING GOF mice.**

(A, C) Immunophenotyping of splenic T cells from STING GOF mice and their WT littermate controls by flow cytometry. (A) Representative gating strategy of naive (CD44^low CD62L^+), central memory (Cen mem, CD44^high CD62L^+) and effector memory (Eff mem, CD44^+ CD62L^-) for both CD4^+ and CD8^+ T cells from STING GOF mice and their WT littermate controls. (B) Ratio of CD3 mean fluorescence intensity (MFI) on splenic naive, central memory and effector memory CD4^+ or CD8^+ T cells from STING GOF mice and their WT littermate controls. Ratio was normalized on the mean of WT controls of each analysis. (C) Proportion of IL-7Rα-expressing cells among splenic naive, central memory and effector memory CD4^+ or CD8^+ T cells from STING GOF mice and their WT littermate controls. Each data point corresponds to one mouse; mean ± SEM are shown per population for four (B) to nine (C) mice from two (B) and three (C) independent experiments (biological replicates). Statistical significances are calculated with a two-tailed Mann–Whitney test.

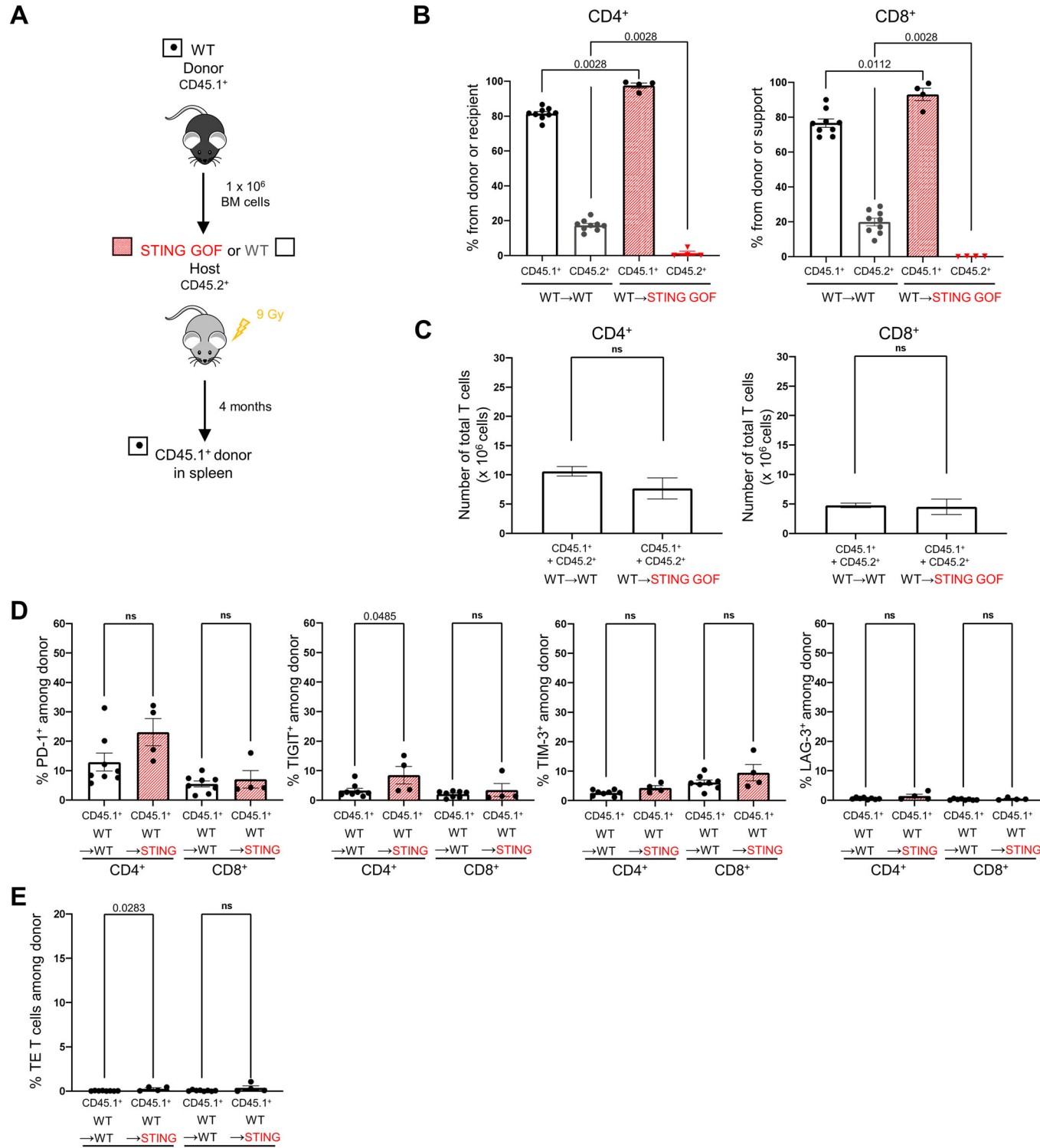

◀ **Figure EV6.   STING GOF radioresistant stroma is not sufficient to induce T cell exhaustion.**

(A–E) STING GOF mice and their WT littermate controls (CD45.2$^+$) were lethally irradiated and then reconstituted with WT BM donor cells (CD45.1$^+$) to generate WT→STING GOF and WT→WT mice. Four months later, spleens were assessed for T cell reconstitution and immunophenotyping. (A) Strategy of WT BM cells transplantations into STING GOF or WT irradiated recipient mice. (B) Proportion of cells derived from WT BM donor cells (CD45.1$^+$) or host cells (CD45.2$^+$) among splenic CD4$^+$ or CD8$^+$ T cells from WT→STING GOF and WT→WT mice. (C) Absolute numbers of total (CD45.1$^+$ and CD45.2$^+$) splenic CD4$^+$ or CD8$^+$ T cells from WT→STING GOF and WT→WT mice. (D) Proportion of PD-1-, TIGIT-, TIM-3-, and LAG-3-expressing cells among splenic CD4$^+$ or CD8$^+$ T cells derived from WT BM donor cells (CD45.1$^+$) from WT→STING GOF and WT→WT mice. (E) Proportion of terminally exhausted (TE) T cells among splenic CD4$^+$ or CD8$^+$ T cells derived from WT BM donor cells (CD45.1$^+$) from WT→STING GOF and WT→WT mice. Each data point corresponds to one mouse; mean ± SEM are shown per population for four to nine mice from two independent experiments (biological replicates). Statistical significances are calculated with a two-tailed Mann–Whitney test; ns (non-significant), *P* > 0.05.

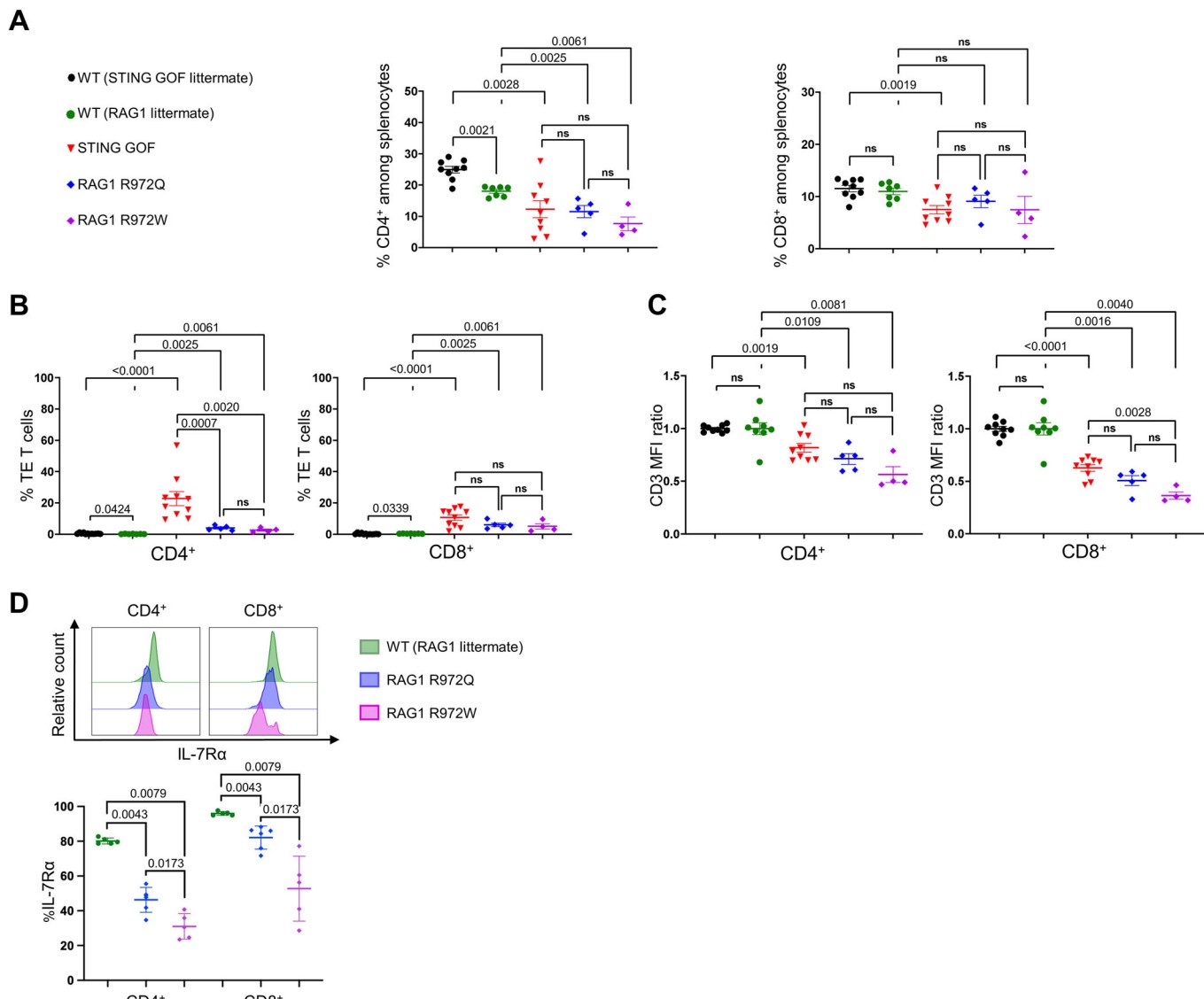

**Figure EV7. Mice carrying *Rag1* hypomorphic mutations also display lymphopenia-associated T cell exhaustion.**

(A–C) Immunophenotyping of T cells from the spleen of hypomorphic RAG1 R972Q and R972W mice and their WT littermate controls by flow cytometry. Data were compared with the previous result obtained from STING GOF mice and their WT littermate controls (Figs. 1F and 3B). (A) Proportion of splenic CD4+ or CD8+ T cells from hypomorphic RAG1 mice and their WT littermate controls. (B) Proportion of terminally exhausted (TE) T cells among splenic CD4+ or CD8+ T cells from hypomorphic RAG1 mice and their WT littermate controls. (C) Ratio of CD3 mean fluorescence intensity (MFI) on splenic CD4+ or CD8+ T cells from hypomorphic RAG1 mice and their WT littermate controls. Ratio was normalized on the mean of WT controls of each analysis. (D) Proportion of IL-7Rα-expressing cells among splenic total CD4+ or CD8 + T cells from hypomorphic RAG1 mice and their WT littermate controls. Representative histograms are shown. Each data point corresponds to one mouse; mean ± SEM are shown per population for four to ten mice from two independent experiments (biological replicates). Statistical significances are calculated with a two-tailed Mann–Whitney test; ns (non-significant), $P > 0.05$.

