## [Peer Review File · EMBO Molecular Medicine]

Lymphopenia drives T cell exhaustion in immunodeficient STING gain-of-function mice

Damien Freytag, Stephane Giorgiutti, Grégoire Hopsomer, Nadege Wadier, Sabine Depauw, Philippe Mertz, Fabrice Augé, Raphael Carapito, Isabelle Couillin, Anne-Sophie Korganow, Francesca Pala, Marita Bosticardo, Luigi Notarangelo, Frederic Rieux-Laucat, Nicolas Riteau, Peggy Kirstetter, and Pauline Soulas-Sprauel

Corresponding author: Pauline Soulas-Sprauel (soulaspa@unistra.fr)

Review Timeline:

Transferred from Review Commons:	6th Feb 25
Editorial Decision:	10th Feb 25
Revision Received:	23rd May 25
Editorial Decision:	1st Jul 25
Revision Received:	23rd Jul 25
Accepted:	25th Jul 25

Editor: Lise Roth

Transaction Report:

Review
COMMONS

This manuscript was transferred to EMBO Molecular Medicine following peer review at Review Commons.

Review #1**1. Evidence, reproducibility and clarity:****Evidence, reproducibility and clarity (Required)******Summary****

In this study, Freytag et al report their findings that mice with a gain-of-function (GOF) mutation in the gene coding for STING suffer from generalized T cell exhaustion characterized by expression of a number of genes known to be associated with exhaustion as well as clear abundance of the surface inhibitory receptors PD-1, TIM-3, TIGIT and LAG-3. This was type I interferon independent. They then show that one of the exhaustion hallmarks, the activation of the NFAT pathway, is due to direct involvement of active STING. However, STING activation was not enough to induce inhibitory receptors. The authors then show that T cells in these mice express lower levels of CD3 and IL-7Ra and suggest that this is related to the exhausted phenotype. Finally, the authors provide evidence suggesting that the exhaustion phenotype in STING-GOF mice is not due to a T cell intrinsic defect or due to the effects of STING-GOF radioresistant cells, but rather due to the lymphopenia that is a feature of these mice.

This is a well-written and concise study with clear results and statistical analyses. There are, however, some issues that need to be addressed experimentally in order to substantiate some of the conclusions.

****Major comments****

Fig 1

- Is expression of inhibitory receptors restricted to EM, CM or naïve T cells? This will nicely reinforce the data in Fig S1 whereby thymic T cells don't seem to be exhausted yet.
- In Fig 1E it appears that even WT T cells are mostly PE. One would expect many more TCF-1^{neg}TIM-3^{neg} cells in WT mice. Is this due to a specific gating strategy? How can it be explained?

Fig S2

As the authors only show FACS plots for the IFNARko cells, it is important to specify whether the statistical analysis and the graph representations in Fig S2A are based on the

same data they show in Fig 1 or they performed parallel experiments with WT, WT-IFNARko, STING-GOF, STING-GOF-IFNARko mice.

Fig 2

NFATc1 levels are now not significant as in in Fig S3B. Is this due to number of experiments/mice? I would suggest to repeat so as to test if the result in S3B is reproducible and to also to show FACS plots.

Fig 3

- In Fig 3A the authors show reduced levels of CD3, which agrees with impaired in vitro responses with aCD3 in their previous study (Buis et al, 2018, JACI). The authors assume that this is due to "consistent TCR stimulation". However, there is no evidence for this. It is plausible that CD3 downregulation is a consequence of exhaustion or that there is less gene expression (their RNAseq analysis could answer this). Do exhausted T cells in the Rag1 hypomorphic model also have reduced CD3?

- I am not sure I understand the data in Fig 3C. Isn't this simply because they are dividing with a smaller number, since STING GOF mice have much fewer cells? They need to show cDC numbers. Also, what does number of cDCs per CD4 T cell means? It is usually one cDC associating with multiple T cells, not the other way around. They could perform microscopy if necessary to investigate number of T cells associating per cDC or duration of cDC-T cell contacts, but I don't think the current analysis/data suggests enhanced TCR signaling. In this regard, I think the discussion points about the role of DCs need to be revised.

- In Fig 3D the authors show a remarkable downregulation of IL-7Ra. Although IL-7 does rapidly downregulate IL-7Ra this is usually seen after in vitro stimulation with rather high doses of recombinant IL-7. Do these mice produce very high levels of IL-7, which could help explain the observed downregulation of IL-7Ra? Or is this also a consequence of exhaustion? In this regard, do T cells in the Rag1 hypomorphic model also have reduced IL-7Ra? Is IL-7Ra gene expression normal in STING-GOF T cells?

Besides homeostatic proliferation, IL-7 through STAT5 induces Bcl2 and survival while it is a potent inducer of surface lymphotoxin. Do STING-GOF T cells have high Bcl2 or lymphotoxin expression? This would show, indirectly at least, that there is active IL-7-IL-7R signaling.

Discussion

- Lines 355-356: the authors state that cDCs are radioresistant. I am not sure that this is correct. In my experience, cDCs are easily "wiped out" by lethal irradiation and there are numerous papers using BM chimeras as a technique to test cDC-intrinsic biology. References or experimental evidence to substantiate this statement is important.

****Minor comments:**** important issues that can confidently get addressed

- Line 147: it is worth mentioning here that downregulation of AP-1 transcription factors is a hallmark of T cell exhaustion, at least in the context of viral infection (see Wherry et al, 2007, Immunity)

- In Fig 1, z-score scale bars are missing color code

- In Fig 2 definitely change the DMXAA color from orange to something more obvious next to the red STING-GPF (maybe blue?)

- Any potential mechanisms as to how STING activation induces Ca²⁺ signaling and NFAT activation? Could it be TBK1-IRF3 or TBK1-NFkB or IFNAR dependent?

- Related to Fig S3. In lines 200-201 the authors state that the data in Fig S3 suggest "that Ca²⁺-NFAT signaling pathway in STING GOF T cells participates to the induction of their exhaustion". However, this is only a correlation. It could be that these observations are the consequence of exhaustion and not the reason for exhaustion. As it stands, the statement cannot be substantiated and therefore I would suggest that the authors change it.

- In their previous publication (Buis et al, 2018, JACI) show that approximately 45% of STING-GOF mice get lung disease. It would be interesting if there was any correlation between the degree of exhaustion and disease severity.

- I don't think the connection between TCR, IL-7Ra, lymphopenia and exhaustion is clear. Could the authors please elaborate potential mechanisms? Especially for IL-7Ra.

- Line 290: there is a "had" missing between "mice" and "similar".

2. Significance:

Significance (Required)

Strengths and advances

The study is novel, the experiments thoroughly performed and I think it addresses a very important question: mechanisms and causes of T cell exhaustion. As the authors rightly state and cite, these data could have important future implications in hematopoietic cell transplantation but I think also in patients receiving immunosuppression and potentially in CAR T therapy for T cell malignancies. In addition, the mouse models could be invaluable for elucidating mechanisms of exhaustion or testing treatments. It is also very encouraging

to see the correct statistical tests performed.

It is true, and cited by the authors, that the association between lymphopenia and T cell exhaustion has been shown before. However, to the best of my knowledge, there are currently no models or mechanistic insights and I think the current study is a promising new advance.

Limitations

I do not find any major weaknesses in this study, with the exception of some overinterpretations and some experiments that can be easily performed to substantiate the findings.

Audience

The study will be relevant for basic and clinical immunologists.

Expertise

My area of expertise is T cell biology. A key focus of my lab is checkpoint receptors in the context of cancer immunotherapy and signaling pathways that co-regulate development and inflammation.

3. How much time do you estimate the authors will need to complete the suggested revisions:

Estimated time to Complete Revisions (Required)

(Decision Recommendation)

Between 1 and 3 months

No

Review #2

1. Evidence, reproducibility and clarity:

Evidence, reproducibility and clarity (Required)

Using a mouse model bearing a gain of function mutation of STING (V154M) the authors report that, in a lymphopenia context, the remaining splenic and lung CD4 and CD8 T cells present an exhausted phenotype (by RNAseq and Flow cytometry) starting very early in life. The authors showed that this phenotype is IFN-Type I independent. Interestingly their results suggest that T cell exhaustion in this model might be a consequence of the lymphopenia. Accordingly, two other mouse models with lymphopenia present enrichment of T cells positive for exhaustion markers, albeit to a lesser level.

This is a well conducted study that took advantage of a mouse model developed and characterized by the team. The experiments are adequately replicated and all required controls were included. The statistical analyses are generally appropriate. It is well written and the figures are clear.

****Major comments:****

1. From RNAseq analysis and flow cytometry detection of surface markers, the authors conclude that the remaining splenic and lung CD4 and CD8 T cells in the STING GOF mouse model are exhausted. This fits with previous observation of reduced proliferation and increased cell death of T cells in this mouse model. A direct functional validation would however be an added value. The authors could quantify by flow cytometry the production of effector cytokines (eg. IL2, Granzyme, IFN-g) by T cells to confirm the increased frequency of exhausted T cells in their model. Moreover, if as proposed in the discussion, exhaustion is actually protective in the case of STING GOF, it would be pivotal to unravel whether phenotypically exhausted and non-exhausted T cells have distinct cytokine secretion abilities in this context.
2. Treating WT cells with an agonist of STING reproduced the increased basal Ca²⁺ level and upregulation of NFATc1 observed in STING GOF bearing T cells. However, this treatment had no effect on the expression of PD-1 and TIM-3. The authors conclude that STING activation is thus not sufficient to induce exhaustion. Although the chimera experiments indeed argue for an environmental driver for T cell exhaustion in this model, it can be argued that 24h of stimulation is not long enough to induce sufficient chronic activation to lead to exhaustion. This conclusion should either be dampened or a longer treatment with the STING agonist should be performed.

3. The authors observed a decreased surface expression of IL7Ra on total CD4 and CD8 T cells. It is further showed for naïve but not for the memory T cell subsets. As IL7 and IL15 are important for homeostatic proliferation of memory CD8 T cells (JT. Tan et al JEM 2002) IL7ra and IL15ra expression on memory T cells (by flow cytometry) should be assessed to support the author conclusion. Moreover, dosing both cytokines in serum and spleen supernatant (by ELISA for example) would support the proposed mechanism.

The proposed experiments will require mouse cohorts (n=8), antibodies for flow cytometry and eventually 2 ELISA kits. Providing mice are available, each experiment should take around a week to be performed and analyzed.

****Minor comments:****

- Although the model has already been described it will help the reader to show the total number of CD4 and CD8 T cells (figure 1, S1, 5 and S5), particularly if the cDC/T cell ratio is one of the driving force behind exhaustion.
- Figure 1: the color code for the z-score is missing
- Figure 1 and 3: the FDR should also be indicated for the GSEA
- Figure legends: the actual number of independent experiments should be indicated rather than "multiple".

2. Significance:

Significance (Required)

This study is conceptually important as it suggests that lymphopenia may promote and aggravate T cell exhaustion. The mechanisms at play are not clear but may be related to the availability of cytokines like IL7 and enhanced T cell activation through increased cDC/T cell ratio.

These results may also have clinical consequences for the standard of care of patients presenting STING GOF mutations and more largely presenting severe lymphopenia. Indeed, T cell exhaustion might be an aggravating feature in case of cancer development in these patients.

This study is likely to be of interest for clinical immunologists and scientists working in the fields of immunodeficiency and cell exhaustion.

3. How much time do you estimate the authors will need to complete the suggested revisions:

Estimated time to Complete Revisions (Required)

(Decision Recommendation)

Between 3 and 6 months

Yes

Revision Plan

Manuscript number: RC-2024-02769

Corresponding author(s): Pauline, Soulas-Sprael

[The “revision plan” should delineate the revisions that authors intend to carry out in response to the points raised by the referees. It also provides the authors with the opportunity to explain their view of the paper and of the referee reports.]

The document is important for the editors of affiliate journals when they make a first decision on the transferred manuscript. It will also be useful to readers of the reprint and help them to obtain a balanced view of the paper.

*If you wish to submit a full revision, please use our "Full Revision" template. **It is important to use the appropriate template to clearly inform the editors of your intentions.**]*

1. General Statements [optional]

This section is optional. Insert here any general statements you wish to make about the goal of the study or about the reviews.

Dear Madam, Dear Sir,

We submit here the preliminary revision and revision plan for our manuscript entitled “Lymphopenia drives T cell exhaustion in immunodeficient STING gain-of-function mice” by Freytag D. et al. We would like to warmly thank the referees for their time, their suggestions which will help to improve the quality of our manuscript, and for their positive comments. In particular, both stated the novelty of the study and its importance, taking into account that our data suggest that lymphopenia is a driver of T cell exhaustion, therefore having clinical impact for patients with lymphopenia in different clinical situations.

*Best regards,
Pr Pauline Soulas-Sprael*

2. Description of the planned revisions

Insert here a point-by-point reply that explains what revisions, additional experimentations and analyses are planned to address the points raised by the referees.

Reviewer #1

Summary

In this study, Freytag et al report their findings that mice with a gain-of-function (GOF) mutation in the gene coding for STING suffer from generalized T cell exhaustion characterized by expression of a number of genes known to be associated with exhaustion as well as clear

Revision Plan

abundance of the surface inhibitory receptors PD-1, TIM-3, TIGIT and LAG-3. This was type I interferon independent. They then show that one of the exhaustion hallmarks, the activation of the NFAT pathway, is due to direct involvement of active STING. However, STING activation was not enough to induce inhibitory receptors. The authors then show that T cells in these mice express lower levels of CD3 and IL-7Ra and suggest that this is related to the exhausted phenotype. Finally, the authors provide evidence suggesting that the exhaustion phenotype in STING-GOF mice is not due to a T cell intrinsic defect or due to the effects of STING-GOF radioresistant cells, but rather due to the lymphopenia that is a feature of these mice.

This is a well-written and concise study with clear results and statistical analyses. There are, however, some issues that need to be addressed experimentally in order to substantiate some of the conclusions.

Major comments

- In Fig 3D the authors show a remarkable downregulation of IL-7Ra. Although IL-7 does rapidly downregulate IL-7Ra this is usually seen after in vitro stimulation with rather high doses of recombinant IL-7. Do these mice produce very high levels of IL-7, which could help explain the observed downregulation of IL-7Ra? Or is this also a consequence of exhaustion? In this regard, do T cells in the Rag1 hypomorphic model also have reduced IL-7Ra? Is IL-7Ra gene expression normal in STING-GOF T cells?

Besides homeostatic proliferation, IL-7 through STAT5 induces Bcl2 and survival while it is a potent inducer of surface lymphotoxin. Do STING-GOF T cells have high Bcl2 or lymphotoxin expression? This would show, indirectly at least, that there is active IL-7-IL-7R signaling.

We propose to perform new experiments on hypomorphic Rag mice to analyse the expression of IL7R α by flow cytometry. Please refer to the 3rd section of this document for the response to the other points of this part.

Reviewer #2:

Using a mouse model bearing a gain of function mutation of STING (V154M) the authors report that, in a lymphopenia context, the remaining splenic and lung CD4 and CD8 T cells present an exhausted phenotype (by RNAseq and Flow cytometry) starting very early in life. The authors showed that this phenotype is IFN-Type I independent. Interestingly their results suggest that T cell exhaustion in this model might be a consequence of the lymphopenia. Accordingly, two other mouse models with lymphopenia present enrichment of T cells positive for exhaustion markers, albeit to a lesser level.

This is a well conducted study that took advantage of a mouse model developed and characterized by the team. The experiments are adequately replicated and all required controls were included. The statistical analyses are generally appropriate. It is well written and the figures are clear.

Major comments:

1) From RNAseq analysis and flow cytometry detection of surface markers, the authors conclude that the remaining splenic and lung CD4 and CD8 T cells in the STING GOF mouse model are exhausted. This fits with previous observation of reduced proliferation and increased cell death of T cells in this mouse model. A direct functional validation would however be an added value. The authors could quantify by flow cytometry the production of effector cytokines (eg. IL2, Granzyme, IFN-g) by T cells to confirm the increased frequency of exhausted T cells in their model. Moreover, if as proposed in the discussion, exhaustion is actually protective in the case of STING GOF, it would be pivotal to unravel whether phenotypically exhausted and non-exhausted T cells have distinct cytokine secretion abilities in this context.

Thanks to the referee for this comment.

It is right that we already showed in our previous paper that proliferation rate of STING GOF T cells is decreased compared to WT control cells, and this somehow a functional defect.

Indeed, we already tried to analyze the production of some cytokines (as IFN γ) by intracellular staining in STING GOF T cells. However, we have encountered some limitations, as the induction of the STING pathway is known to strongly increase the production of IFN γ and induce TH1 pathway (see for example this paper: <https://doi-org.proxy.insermbiblio.inist.fr/10.1136/jitc-2021-003459>). Consequently, it is impossible to assess the functional impact of exhaustion in STING GOF T cells by comparing them to WT cells. Thus, we tried to compare IFN γ expression between exhausted and non exhausted STING GOF T cells to better appreciate functional defects. Because IFN γ staining is too low at basal state to draw any conclusions, we were obliged to stimulate T cells to increase cytokine production. However, after stimulation with anti-CD3 and anti-CD28, the immune checkpoint expression by STING GOF T cells is so high (almost 80-90% taking into account PD1 or TIM3 staining) that it is no longer possible to determine which cells were initially exhausted.

However, we can propose to do a sorting of exhausted and non exhausted CD4⁺ and CD8⁺ T cells from STING GOF mice by flow cytometry, and analyze by qPCR the production of key effector cytokines (IFN γ , IL-2, granzyme). In this way, we can hope to see if there is a difference in cytokine production between the two populations (exhausted, non exhausted) in STING GOF context. We think this is the best way to reply to this question in the face of the increase production of IFN γ induced by STING pathway.

Revision Plan

3. Description of the revisions that have already been incorporated in the transferred manuscript

Please insert a point-by-point reply describing the revisions that were already carried out and included in the transferred manuscript. If no revisions have been carried out yet, please leave this section empty.

*Please note that the modifications inserted into the text of the revised manuscript are in red, in the version of the manuscript entitled “Soulas-Sprauel P and coll STING GOF mouse model_revision_def_Manuscript with tracked changes” and “Soulas-Sprauel P and coll STING GOF mouse model_SupportingInformation_revision_def_Manuscript with tracked changes”, uploaded as **Manuscript related files**.*

We have shortened the abstract to 175 words according to the EMBO Molecular medicine journal guidelines.

Reviewer #1 (continued) **Major comments**

Fig 1

- Is expression of inhibitory receptors restricted to EM, CM or naïve T cells? This will nicely reinforce the data in Fig S1 whereby thymic T cells don't seem to be exhausted yet.

*Thank you for this interesting question which will help to understand the chronology of T cell exhaustion in our model. To answer this question, we performed a phenotyping experiment (by flow cytometry) on STING GOF and control mice (n=4 mice per group) with the analysis of expression of the 4 inhibitory receptors (PD-1, TIM-3, LAG-3, TIGIT) on splenic naïve, effector memory (EM) and central memory (CM) CD4⁺ and CD8⁺ T cells (**New Figure S1**).*

*The results confirmed the increased expression of these 4 inhibitory receptors on total T cells in STING GOF mice (**Figure S1A**), and show a moderate increase of inhibitory receptors expression at naïve stage in STING GOF T cells (in particular TIM-3, marker of terminal exhaustion, was not increased at naïve stage) with worsening at memory stage (**Figure S1B**). This was added in the results section **lines 164-174**.*

- In Fig 1E it appears that even WT T cells are mostly PE. One would expect many more TCF-1^{neg}TIM-3^{neg} cells in WT mice. Is this due to a specific gating strategy? How can it be explained?

We apologize for this misunderstanding. Indeed, the FACS plots in Figure 1E are gated on CD4⁺PD-1⁺ and CD8⁺PD-1⁺ cells (as usually done in the literature in order to analyse the proportions of PE and TE cells with the additional TCF-1 and TIM-3 stainings), as indicated above the plots on the left figure. In addition, the figure on the right present the percentages of TE T cells

Revision Plan

*among total T cells. In order to increase the clarity of these 2 figures, we have separated them in **Figures 1E and F**, respectively, and modified the **legend of Figure 1**.*

Fig S2

As the authors only show FACS plots for the IFNARko cells, it is important to specify whether the statistical analysis and the graph representations in Fig S2A are based on the same data they show in Fig 1 or they performed parallel experiments with WT, WT-IFNARko, STING-GOF, STING-GOF-IFNARko mice.

*Yes, as mentioned by the referee, statistical analysis and the graph representations in Figure S2A are based on the same data showed in Figure 1 for STING GOF mice. This has been more precisely written in the **legend of Figure S2**.*

Fig 2

NFATc1 levels are now not significant as in in Fig S3B. Is this due to number of experiments/mice? I would suggest to repeat so as to test if the result in S3B is reproducible and to also to show FACS plots.

*Thank you for this relevant comment. In order to reply to this point, we have done and added in the **Figure 2** additional mice (n=2), in order to have n=6 mice per group. With these new data, we show that NFATc1 is increased in STING GOF non treated T cells (red), as well as in WT DMXAA-treated T cells (light blue, compared to WT non treated T cells (black), in a statistically significant manner, after 24h of culture in vitro, which is more coherent with the ex vivo results (**Figure S4B**) and with our conclusions.*

Fig 3

- In Fig 3A the authors show reduced levels of CD3, which agrees with impaired in vitro responses with aCD3 in their previous study (Buis et al, 2018, JACI). The authors assume that this is due to "consistent TCR stimulation". However, there is no evidence for this. It is plausible that CD3 downregulation is a consequence of exhaustion or that there is less gene expression (their RNAseq analysis could answer this). Do exhausted T cells in the Rag1 hypomorphic model also have reduced CD3?

*As mentioned by the referee, the decrease of CD3 expression is indeed consistent with the significant decreased response of STING GOF T cells to anti-CD3 stimulation in vitro in our previous publication. We have added this point in the results part, **lines 232-233**. As explained before, STING GOF CD4⁺ and CD8⁺ naïve T cells seem to be only moderately concerned by the exhausted phenotype. It allows us to hypothesize that exhaustion starts to set up after TCR (and also IL7R, see further points) activation, because the downregulation of expression of IL7R α and CD3 already takes place on naïve T cells in these mice (new data added for CD3 expression in **Figure S5**), and in the text of results section **lines 236-239**. In addition, CD3 downregulation was*

Revision Plan

*also detected in Rag hypomorphic mice, which also display T cell exhaustion. These results have been added in a **new Figure S7C**, with modified legend, and in the text of the results, **lines 328-330**.*

*Consistent with the internalization of the TCR complex upon its stimulation, the down regulation of CD3 expression was only detectable at the protein level, since mRNA levels were not diminished in our RNAseq analysis. This has been added in the text results **lines 234-236** and in **Table S2**.*

*Finally, we showed a significant engagement of TCR in STING GOF T cells compared to T cells from control mice using four GSEA signatures. This was added in the text of the results **lines 229-230** and in **Table S1**. In addition, we would like to inform the referee that the GSEA TCR signature initially used in the manuscript in Fig. 3A (WP T Cell Antigen Receptor TCR Signaling Pathway) was not available anymore in GSEA application at the date of last consultation (Januray, 10th, 2025). Therefore we propose to remove it from the paper. Then we have replaced the signature in Figure 3A by the "PID-TCR-PATHWAY" signature.*

Altogether, these data strongly suggest that there is a consistent TCR stimulation in STING GOF T cells, which seems to precede the onset of T cell exhaustion.

Fig 3

- I am not sure I understand the data in Fig 3C. Isn't this simply because they are dividing with a smaller number, since STING GOF mice have much fewer cells? They need to show cDC numbers. Also, what does number of cDCs per CD4 T cell means? It is usually one cDC associating with multiple T cells, not the other way around. They could perform microscopy if necessary to investigate number of T cells associating per cDC or duration of cDC-T cell contacts, but I don't think the current analysis/data suggests enhanced TCR signaling. In this regard, I think the discussion points about the role of DCs need to be revised.

*We thank the referee for this question about cDCs numbers. Indeed, this increased ratio of cDC versus T cells in STING GOF mice is really linked to a decreased number of T cells. The numbers of splenic cDCs is not modified in STING GOF mice compared to control mice. These data (numbers of T cells and cDCs) have been added in the text of the results **lines 246-254**. The model we propose here is therefore the following: instead of saying that one T cell will associate with multiple cDCs, we propose that the very few STING GOF T cells have more chance to receive activation signal through TCR-MHC complex, mainly provided by cDC whose number is normal. The text has been modified to this effect (**lines 254-256**). We have also revised the discussion by removing the sentence "Of note, the observation that the lymphopenia leads to a higher number of cDCs per CD4⁺ T cells than for CD8⁺ T cells may help explain the more severe T cell exhaustion phenotype observed for CD4⁺ T cells" (**lines 371-373**). About the comment in enhanced TCR signaling, please the reply to the other comment above.*

- In Fig 3D the authors show a remarkable downregulation of IL-7Ra. Although IL-7 does rapidly downregulate IL-7Ra this is usually seen after in vitro stimulation with rather high doses of recombinant IL-7. Do these mice produce very high levels of IL-7, which could help explain the observed downregulation of IL-7Ra? Or is this also a consequence of exhaustion? In this regard,

Revision Plan

do T cells in the Rag1 hypomorphic model also have reduced IL-7Ra? Is IL-7Ra gene expression normal in STING-GOF T cells?

Besides homeostatic proliferation, IL-7 through STAT5 induces Bcl2 and survival while it is a potent inducer of surface lymphotoxin. Do STING-GOF T cells have high Bcl2 or lymphotoxin expression? This would show, indirectly at least, that there is active IL-7-IL-7R signaling.

We replied partly to these comments in the 2nd section of this revision plan, about the fact that we will perform new experiments on hypomorphic Rag mice to analyse expression of IL7R α .

Apart from this point, please find below our responses to the other comments:

- *We also observed a significant decrease of mRNA encoding IL7R α in our RNA seq analysis in CD4⁺ and CD8⁺ T cells. We have added these data in the text of the results **lines 263-269** and in a **new Table S2**. Therefore, it is an additional data in favor of IL-7R activation. Indeed, according to the literature (see for example this review: <https://doi.org/10.1016/j.cyto.2022.156049> and in the references added in the results section **line 269, references 32-34**), the engagement of IL-7R by IL-7 downregulates IL7R α expression both at the transcriptional and protein level, and both are seen on STING GOF T cells.*
- *Other data are also in accordance with the engagement of IL-7R in STING GOF T cells:*
 - o *bcl2 mRNA expression is increased in CD4⁺ T cells in STING GOF mice, and also in STING GOF CD8⁺ T cells in a statistical manner.*
 - o *The mRNA encoding p27^{kip1} (cdkn1b gene), a member of the universal cyclin-dependent kinase inhibitor (CDKI) family, is significantly decreased in STING GOF CD4⁺ and CD8⁺ T cells in our RNA-seq analysis. It is well described in the literature that this gene is known to be negatively regulated after IL-7R stimulation.*
 - o *Another gene known to be regulated by IL-7R activation is gfi1, encoding GF11 (growth-factor independent 1) which is induced by IL-7 stimulation to repress IL-7R α expression. In our RNA-seq analysis, we observed a significant decreased of gfi1 mRNA both in CD4⁺ and CD8⁺ T cells in STING GOF mice compared to control mice.*
 - o *Finally, SOCS-1 (suppressor of the cytokine signaling 1) is another target gene of IL-7R engagement, which is upregulated after stimulation with IL-7, to negatively regulate IL-7R signaling pathways. We observed an increased expression of mRNA encoding SOCS-1 in STING GOF CD4⁺ and CD8⁺ T cells compared to control cells, in the RNA-seq analysis.*
 - o *We added these data in the text of the results **lines 263-272**, with three references of reviews about IL-7 biology, and in the new **Table S2**.*
- *Concerning the potential causes of IL-7R increased stimulation in STING GOF T cells:*
 - o *We previously did an experiment to measure serum IL-7 level in STING GOF mice, with ELISA kit (ref Invitrogen Mouse IL-7 ELISA kit EMIL7). Unfortunately, although the standard curve was correct, we didn't manage to detect any IL-7 in the serum, even in low-diluted serum (dilution 1/2), in control and STING GOF mice. The assay range is announced by the supplier as 8.23-*

Revision Plan

6000 pg/mL. Because we didn't measure any detectable level of IL-7 in serum of STING GOF mice, we can suppose that, if increased, is it not sufficiently increased to be detectable by ELISA kits. As mentioned in the literature on IL-7, serum IL-7 levels in mice are very low and difficult to detect by ELISA or immunochemistry (references 32-33). Indeed, the mean level of IL-7 in serum in normal mice is described in the literature as being less than 10 pg/ml, depending on the references (see for example Figure 5 in this reference: <http://www.biomedcentral.com/1471-2407/10/12>). In addition, it is described that the amount of IL-7 is regulated more by consumption than by production, and that the response of cells to IL-7 is more regulated by IL-R α expression than by the production of IL-7 itself (references 32-33).

- We can hypothesize that T cells in the situation of T cell lymphopenia in STING GOF mice are more activable by IL-7 because they will encounter an abundant amount of IL-7 per T cell, compared to a physiological situation, as described in lymphopenia-induced proliferation. This has been added in the results section, **lines 269-272**.

- Finally, in STING GOF mice, IL7-R α expression is already decreased at the stage of naïve T cells (**Figure S5**) where exhaustion seems to be only moderate (**new Figure S1**). Therefore, it allows us to propose that IL-7R stimulation, which is surely induced by lymphopenia, as the TCR stimulation, precedes exhaustion and is not only a consequence. The discussion part has been completed with this model proposal in **lines 376-383**.

Discussion

- Lines 355-356: the authors state that cDCs are radioresistant. I am not sure that this is correct. In my experience, cDCs are easily "wiped out" by lethal irradiation and there are numerous papers using BM chimeras as a technique to test cDC-intrinsic biology. References or experimental evidence to substantiate this statement is important.

*Thanks to the referee for this comment. The referee is right, this was a mistake, we apologize for this. We have removed the following part of the sentence about cDC in the discussion **line 402**: "which are also included in the radioresistant cell population".*

Minor comments:

- Line 147: it is worth mentioning here that downregulation of AP-1 transcription factors is a hallmark of T cell exhaustion, at least in the context of viral infection (see Wherry et al, 2007, Immunity).

*We have modified the sentence in the results section **line 146** in order to strengthen this point.*

- In Fig 1, z-score scale bars are missing color code.

Revision Plan

There must have been an error when formatting the manuscript. We hope that this is now corrected. In case the error is still present on your copy, please find below the color scale.

Z-score

- In Fig 2 definitely change the DMXAA color from orange to something more obvious next to the red STING-GPF (maybe blue?).

*We have changed the color of the orange points, to a light blue color, for the entire **Figure 2**.*

- Any potential mechanisms as to how STING activation induces Ca²⁺ signaling and NFAT activation? Could it be TBK1-IRF3 or TBK1-NFκB or IFNAR dependent?

*A negative regulation of STING by STIM1 has been described in the literature (Srikanth S and coll, Nat Immunol, 2019, 20(2), 152-162): in resting state in WT cells, STIM1 retains STING at the ER. When STING is activated (via cGAMP or SAVI gain of function mutations), the STING–STIM1 interaction is disrupted, which allows STING to exit the ER, moves to the ERGIC, and begin trafficking and signaling. We can therefore hypothesize that there could be a negative inter-regulation of STING and STIM1 at the ER: in case of STING activation (as in our STING GOF model or after DMXAA treatment in WT cells), STIM1 is not retained by STING at the ER anymore and could move to the ER-PM (plasma membrane) junction to operate SOCE (store-operated Ca²⁺ entry) and allow entry of calcium into the cell, then activating NFAT pathway. This hypothesis has been added in the text of the discussion **lines 392-396**.*

*In addition, the NFAT activation upon DMXAA treatment was already described to be type I IFN-independent, as it was already indicated in the discussion part **lines 390-392** (and reference 40).*

- Related to Fig S3. In lines 200-201 the authors state that the data in Fig S3 suggest "that Ca²⁺-NFAT signaling pathway in STING GOF T cells participates to the induction of their exhaustion". However, this is only a correlation. It could be that these observations are the consequence of exhaustion and not the reason for exhaustion. As it stands, the statement cannot be substantiated and therefore I would suggest that the authors change it.

*We replaced "participates" by "could be associated with" **lines 207-208** in the text of the results.*

Revision Plan

- In their previous publication (Buis et al, 2018, JACI) show that approximately 45% of STING-GOF mice get lung disease. It would be interesting if there was any correlation between the degree of exhaustion and disease severity.

We have never analyzed T cell exhaustion in blood, lymphoid organs or lungs in parallel with lung disease in the same mice. However, while lung disease in mice can be heterogeneous (as the 45% of disease described in our previous paper), T cell exhaustion is present in all the mice, as can be seen in Figure 1. Thus, at this stage, we cannot conclude for a direct correlation between lower potential exhaustion and a more severe disease. In addition, other factors are surely implicated in the severity of the disease. We have added a sentence to this effect in the discussion lines 350-352 for the sake of clarity.

- I don't think the connection between TCR, IL-7Ra, lymphopenia and exhaustion is clear. Could the authors please elaborate potential mechanisms? Especially for IL-7Ra.

We hope that the referee is more convinced with the new experiments added in the manuscript (see the reply to other points in this revision plan), and the sentences added in the new discussion part in this point lines 376-383.

- Line 290: there is a "had" missing between "mice" and "similar".

We have corrected it.

Strengths and advances

The study is novel, the experiments thoroughly performed and I think it addresses a very important question: mechanisms and causes of T cell exhaustion. As the authors rightly state and cite, these data could have important future implications in hematopoietic cell transplantation but I think also **in patients receiving immunosuppression and potentially in CAR T therapy for T cell malignancies**. In addition, the mouse models could be invaluable for elucidating mechanisms of exhaustion or testing treatments. It is also very encouraging to see the correct statistical tests performed. It is true, and cited by the authors, that the association between lymphopenia and T cell exhaustion has been shown before. However, to the best of my knowledge, there are currently no models or mechanistic insights and I think the current study is a promising new advance.

We have added this point (in red) in the discussion line 423, because we do agree with the referee that it would not only concern hematopoietic cell transplantation (HCT) clinical situation, and deserves to be added in the discussion.

Limitations

I do not find any major weaknesses in this study, with the exception of some overinterpretations and some experiments that can be easily performed to substantiate the findings.

Revision Plan

Audience

The study will be relevant for basic and clinical immunologists.

Expertise

My area of expertise is T cell biology. A key focus of my lab is checkpoint receptors in the context of cancer immunotherapy and signaling pathways that co-regulate development and inflammation.

Reviewer #2 (continued)

Major comments:

3) The authors observed a decreased surface expression of IL7Ra on total CD4 and CD8 T cells. It is further showed for naïve but not for the memory T cell subsets. As IL7 and IL15 are important for homeostatic proliferation of memory CD8 T cells (JT. Tan et al JEM 2002) IL7ra and IL15ra expression on memory T cells (by flow cytometry) should be assessed to support the author conclusion. Moreover, dosing both cytokines in serum and spleen supernatant (by ELISA for example) would support the proposed mechanism.

*We added the data for memory T cells: our data show that memory T cells in STING GOF mice display a high decrease of IL-7R α expression compared to T cells from control mice, as do naïve T cells. These data were added in **Figure S5** and modified the text in the results section **lines 260-263**.*

For the dosage of serum IL-7, please refer to the reply made to referee 1 in section 3, and also the other points about IL-7R pathway.

About the comment on IL-15, please refer to section 4.

Minor comments:

- Although the model has already been described it will help the reader to show the total number of CD4 and CD8 T cells (figure 1, S1, 5 and S5), particularly if the cDC/T cell ratio is one of the driving force behind exhaustion.

*Thanks to the referee for suggesting to mention the number of CD4⁺ and CD8⁺ T cells in STING GOF mice. These have been added to the text of Results section in relation to cDCs numbers, **lines 251-254**. See also the reply made to another comment raised by referee 1.*

- Figure 1: the color code for the z-score is missing

As mentioned in this section for the minor comments of referee 1, there must have been an error when formatting the manuscript. We hope that this is now corrected. In case the error is still present on your copy, please find below the color scale.

Revision Plan

- Figure 1 and 3: the FDR should also be indicated for the GSEA

FDR have been added for all GSEA signatures presented.

- Figure legends: the actual number of independent experiments should be indicated rather than "multiple".

The precise number of independent experiments have been added in each legend. In addition, the number of analyses has been updated for Figure 2 which incorporates new data in the revised version of the manuscript.

Reviewer #2 (Significance):

This study is conceptually important as it suggests that lymphopenia may promote and aggravate T cell exhaustion. The mechanisms at play are not clear but may be related to the availability of cytokines like IL7 and enhanced T cell activation through increased cDC/T cell ratio. These results may also have clinical consequences for the standard of care of patients presenting STING GOF mutations and more largely presenting severe lymphopenia. Indeed, T cell exhaustion might be an aggravating feature in case of cancer development in these patients. This study is likely to be of interest for clinical immunologists and scientists working in the fields of immunodeficiency and cell exhaustion.

Please note that we have moved up Grégoire Hopsomer's place in the list of authors to 3rd author, in view of his involvement in the experiments carried out to revise the manuscript.

4. Description of analyses that authors prefer not to carry out

Please include a point-by-point response explaining why some of the requested data or additional analyses might not be necessary or cannot be provided within the scope of a revision. This can be due to time or resource limitations or in case of disagreement about the necessity of such additional data given the scope of the study. Please leave empty if not applicable.

Reviewer #2 (continued)

Major comments:

2) Treating WT cells with an agonist of STING reproduced the increased basal Ca²⁺ level and upregulation of NFATc1 observed in STING GOF bearing T cells. However, this treatment had no effect on the expression of PD-1 and TIM-3. The authors conclude that STING activation is thus not sufficient to induce exhaustion. Although the chimera experiments indeed argue for an environmental driver for T cell exhaustion in this model, it can be argued that 24h of stimulation is not long enough to induce sufficient chronic activation to lead to exhaustion. This conclusion should either be dampened or a longer treatment with the STING agonist should be performed.

*We thank the referee for this remark. In fact, we tried to do some in vitro cultures of T cells from WT mice with DMXAA for 48h (and even 72h), and the mortality of the cells was very high. You can see the results of viability in the figure below for 24h of culture. It is coherent with the literature where it has been described that STING agonist induce a high apoptosis of cells in vitro. Therefore it won't be possible to do the experiment proposed by the referee. We added a sentence in the results section **lines 216-217** to explain this and dampen the conclusion (**line 219**).*

3) The authors observed a decreased surface expression of IL7Ra on total CD4 and CD8 T cells. It is further showed for naïve but not for the memory T cell subsets. As IL7 and IL15 are important for homeostatic proliferation of memory CD8 T cells (JT. Tan et al JEM 2002) IL7ra and IL15ra expression on memory T cells (by flow cytometry) should be assessed to support the author conclusion. Moreover, dosing both cytokines in serum and spleen supernatant (by ELISA for example) would support the proposed mechanism.

Revision Plan

About the remarks on IL-15:

As suggested and explained in section 3, we analyzed the expression of the IL-7R α on memory T-cell populations. These data show that the decrease of IL-7R α expression concerns both naïve and memory STING GOF T cells, compared to their WT controls, thus reinforcing the mechanism of homeostatic proliferation.

We do agree that, similar to IL-7, IL-15 plays a significant role in the homeostatic proliferation, especially for memory T cells. Coherently, our RNA-seq analysis confirms an IL-15 signature (REACTOME-INTERLEUKIN_15_SIGNALING) significantly enriched in STING GOF CD4⁺ (NES=1.85, p <0.001, FDR=0.007) and CD8⁺ (NES=1.56, p =0.014, FDR=0.071) T cells, compared to control T cells. It seems therefore that IL-15R pathway is also engaged.

Although dosing cytokines in serum is of interest, we prefer not to include this kind of analysis for the following reasons. First, these cytokines are so low in serum that quantification is not always possible. As explained in section 3, we didn't manage to detect any IL-7 in the serum of our mice because of its too low concentration (references 32-22). It will be either difficult to use IL-15 serum levels as a marker for homeostatic proliferation for the same reason. Indeed, IL-15 is almost undetectable in biologic fluids (see for example this paper:

[https://doi-org.proxy.insermbiblio.inist.fr/10.1016/s1074-7613\(02\)00429-6](https://doi-org.proxy.insermbiblio.inist.fr/10.1016/s1074-7613(02)00429-6)). As an example, Anderson and coll showed that the mean level of IL-15 in B6 mice is around 40 pg/ml while the detection threshold of murine IL-15 dosage ELISA kit is 125 pg/ml (<https://doi.org/10.1210/en.2015-1746>).

In addition to these technical issues, we believe that these cytokine concentrations, although indicative, are not necessarily representative of T cell stimulation. For example, IL-7 levels are directly regulated more by consumption than by production. In addition, IL-15 seems to mostly operate in a cell contact-dependent manner through the trans-presentation of membrane-bound complexes of IL-15 and IL-15R α on the producing cells to IL-2/IL-15 receptor- β chain (CD122) and γc on the responding cells (<https://doi-org.proxy.insermbiblio.inist.fr/10.4110/in.2024.24.e11>), suggesting that the soluble IL-15 doesn't seem to be the best correlate with potential IL-15R engagement.

Our primary focus in Figure 3, is on the impact of lymphopenia and homeostatic proliferation on naïve T cells, since it appears that these cells are already impacted by TCR and IL-7 stimulation before acquiring the T cell exhaustion phenotype. Therefore, we would prefer to concentrate here on IL-7 since naïve T cells are highly dependent on IL-7 for their survival and homeostatic maintenance, particularly under conditions of lymphopenia. While IL-15 plays an important role in the proliferation and persistence of memory T cells, introducing this additional layer of complexity is less aligned with our current objectives. Focusing exclusively on IL-7 allows us to better understand the specific dynamics affecting naïve T cells in STING GOF mice without complicating the analysis with homeostatic proliferation of memory T cell, which also seems to occur in our mice (GSEA signature of IL-15R engagement can be added in supplemental if suggested by the referee) but which we believe is less connected with T cell exhaustion acquisition.

10th Feb 2025

Dear Prof. Soulas-Sprauel,

Thank you for the submission of your manuscript to our editorial offices. I have now had the opportunity to read it, together with the referees' reports and your rebuttal letter, and to discuss them with the other members of our editorial team.

We agree that the study fits the scope of the journal, and we appreciate that you are willing to address most of the referees' concerns, except for referee #2's points 2 and 3, that should nevertheless be addressed by adequate discussion. We thus encourage you to submit a revised version of your manuscript, including the modifications and revisions described in your point-by-point letter.

Acceptance of the manuscript will entail a second round of review. EMBO Molecular Medicine encourages a single round of revision only and therefore, acceptance or rejection of the manuscript will depend on the completeness of your responses included in the next, final version of the manuscript. For this reason, and to save you from any frustrations in the end, I would strongly advise against returning an incomplete revision.

We require:

- 1) A .docx formatted version of the manuscript text (including legends for main figures, EV figures and tables). Please make sure that the changes are highlighted to be clearly visible.
- 2) Individual production quality figure files as .eps, .tif, .jpg (one file per figure). For guidance, download the 'Figure Guide PDF' (<https://www.embopress.org/page/journal/17574684/authorguide#figureformat>).
- 3) At EMBO Press we ask authors to provide source data for the main figures. Our source data coordinator will contact you to discuss which figure panels we would need source data for and will also provide you with helpful tips on how to upload and organize the files.
- 4) A .docx formatted letter INCLUDING the reviewers' reports and your detailed point-by-point responses to their comments. As part of the EMBO Press transparent editorial process, the point-by-point response is part of the Review Process File (RPF), which will be published alongside your paper.
- 5) A complete author checklist, which you can download from our author guidelines (<https://www.embopress.org/page/journal/17574684/authorguide#submissionofrevisions>). Please insert information in the checklist that is also reflected in the manuscript. The completed author checklist will also be part of the RPF.
- 6) All Materials and Methods need to be described in the main text using our 'Structured Methods' format. According to this format, the Methods section includes a Reagents and Tools Table (listing key reagents, experimental models, software and relevant equipment and including their sources and relevant identifiers) followed by a Methods and Protocols section describing the methods, ideally using a step-by-step protocol format. The aim is to facilitate adoption of the methodologies across labs. Please download and fill our Reagents and Tools Table template (.docx), which you can find in our author guidelines: <https://www.embopress.org/page/journal/14693178/authorguide#structuredmethods>. When submitting your revised manuscript, please do not include the Reagents and Tools Table in the Methods section of the manuscript but upload it as a separate file choosing the file type "Reagent Table". An example of a Method paper with Structured Methods can be found here: <https://www.embopress.org/doi/10.15252/msb.20178071>
- 7) Please note that all corresponding authors are required to supply an ORCID ID for their name upon submission of a revised manuscript.
- 8) It is mandatory to include a 'Data Availability' section after the Materials and Methods. Before submitting your revision, primary datasets produced in this study need to be deposited in an appropriate public database, and the accession numbers and

database listed under 'Data Availability'. Please remember to provide a reviewer password if the datasets are not yet public (see <https://www.embopress.org/page/journal/17574684/authorguide#dataavailability>).

9) For data quantification: please specify the name of the statistical test used to generate error bars and P values, the number (n) of independent experiments (specify technical or biological replicates) underlying each data point and the test used to calculate p-values in each figure legend. The figure legends should contain a basic description of n, P and the test applied. Graphs must include a description of the bars and the error bars (s.d., s.e.m.). Please provide exact p values.

10) Our journal encourages inclusion of *data citations in the reference list* to directly cite datasets that were re-used and obtained from public databases. Data citations in the article text are distinct from normal bibliographical citations and should directly link to the database records from which the data can be accessed. In the main text, data citations are formatted as follows: "Data ref: Smith et al, 2001" or "Data ref: NCBI Sequence Read Archive PRJNA342805, 2017". In the Reference list, data citations must be labeled with "[DATASET]". A data reference must provide the database name, accession number/identifiers and a resolvable link to the landing page from which the data can be accessed at the end of the reference. Further instructions are available at .

11) We replaced Supplementary Information with Expanded View (EV) Figures and Tables that are collapsible/expandable online. A maximum of 5 EV Figures can be typeset. EV Figures should be cited as 'Figure EV1, Figure EV2' etc... in the text and their respective legends should be included in the main text after the legends of regular figures.

12) The paper explained: EMBO Molecular Medicine articles are accompanied by a summary of the articles to emphasize the major findings in the paper and their medical implications for the non-specialist reader. Please provide a draft summary of your article highlighting

13) Author contributions: CRedit has replaced the traditional author contributions section because it offers a systematic machine readable author contributions format that allows for more effective research assessment. Please remove the Authors Contributions from the manuscript and use the free text boxes beneath each contributing author's name in our system to add specific details on the author's contribution. More information is available in our guide to authors.

Please also suggest a visual abstract to illustrate your article as a PNG file 550 px wide x 300-600 px high. A cropped portion of this image will serve as thumbnail for the table of content on our webpage.

16) As part of the EMBO Publications transparent editorial process initiative (see our Editorial at <http://embomolmed.embopress.org/content/2/9/329>), EMBO Molecular Medicine will publish online a Review Process File (RPF) to accompany accepted manuscripts.

In the event of acceptance, this file will be published in conjunction with your paper and will include the anonymous referee reports, your point-by-point response and all pertinent correspondence relating to the manuscript. Let us know whether you

agree with the publication of the RPF and as here, if you want to remove or not any figures from it prior to publication. Please note that the Authors checklist will be published at the end of the RPF.

I look forward to receiving your revised manuscript.

Yours sincerely,

Lise Roth

Rev_Com_number: RC-2024-02769
New_manu_number: EMM-2025-21368-T
Corr_author: Soulas-Sprauel
Title: Lymphopenia drives T cell exhaustion in immunodeficient STING gain-of-function mice

Revision Plan

Manuscript number: RC-2024-02769

Corresponding author(s): Pauline, Soulas-Sprauel

[The “revision plan” should delineate the revisions that authors intend to carry out in response to the points raised by the referees. It also provides the authors with the opportunity to explain their view of the paper and of the referee reports.]

The document is important for the editors of affiliate journals when they make a first decision on the transferred manuscript. It will also be useful to readers of the reprint and help them to obtain a balanced view of the paper.

*If you wish to submit a full revision, please use our "Full Revision" template. **It is important to use the appropriate template to clearly inform the editors of your intentions.**]*

1. General Statements [optional]

This section is optional. Insert here any general statements you wish to make about the goal of the study or about the reviews.

Dear Madam, Dear Sir,

We submit here the point-by-point responses to referees comments for our manuscript entitled “Lymphopenia drives T cell exhaustion in immunodeficient STING gain-of-function mice” by Freytag D. et al. We would like to warmly thank the referees for their time, their suggestions which helped to improve the quality of our manuscript, and for their positive comments. In particular, both stated the novelty of the study and its importance, taking into account that our data suggest that lymphopenia is a driver of T cell exhaustion, therefore having clinical impact for patients with lymphopenia in different clinical situations.

*Best regards,
Pr Pauline Soulas-Sprauel*

2. Description of the revisions that have been incorporated in the transferred manuscript

Please insert a point-by-point reply describing the revisions that were already carried out and included in the transferred manuscript. If no revisions have been carried out yet, please leave this section empty.

Please note that the modifications inserted into the text of the revised manuscript are in red.

Reviewer #1

Revision Plan

Summary

In this study, Freytag et al report their findings that mice with a gain-of-function (GOF) mutation in the gene coding for STING suffer from generalized T cell exhaustion characterized by expression of a number of genes known to be associated with exhaustion as well as clear abundance of the surface inhibitory receptors PD-1, TIM-3, TIGIT and LAG-3. This was type I interferon independent. They then show that one of the exhaustion hallmarks, the activation of the NFAT pathway, is due to direct involvement of active STING. However, STING activation was not enough to induce inhibitory receptors. The authors then show that T cells in these mice express lower levels of CD3 and IL-7Ra and suggest that this is related to the exhausted phenotype. Finally, the authors provide evidence suggesting that the exhaustion phenotype in STING-GOF mice is not due to a T cell intrinsic defect or due to the effects of STING-GOF radioresistant cells, but rather due to the lymphopenia that is a feature of these mice.

This is a well-written and concise study with clear results and statistical analyses. There are, however, some issues that need to be addressed experimentally in order to substantiate some of the conclusions.

Major comments

Fig 1

- Is expression of inhibitory receptors restricted to EM, CM or naïve T cells? This will nicely reinforce the data in Fig S1 whereby thymic T cells don't seem to be exhausted yet.

*Thank you for this interesting question which will help to understand the chronology of T cell exhaustion in our model. To answer this question, we performed a phenotyping experiment (by flow cytometry) on STING GOF and control mice (n=4 mice per group) with the analysis of expression of the 4 inhibitory receptors (PD-1, TIM-3, LAG-3, TIGIT) on splenic naive, effector memory (EM) and central memory (CM) CD4⁺ and CD8⁺ T cells (**New Expanded View (EV) Figure 1**).*

*The results confirmed the increased expression of these 4 inhibitory receptors on total T cells in STING GOF mice (**Expanded View Figure 1A**), and show a moderate increase of inhibitory receptors expression at naive stage in STING GOF T cells (in particular TIM-3, marker of terminal exhaustion, was not increased at naive stage) with worsening at memory stage (**Expanded View Figure 1B**). This was added in the results section **lines 187-195**.*

- In Fig 1E it appears that even WT T cells are mostly PE. One would expect many more TCF-1^{neg}TIM-3^{neg} cells in WT mice. Is this due to a specific gating strategy? How can it be explained?

We apologize for this misunderstanding. Indeed, the FACS plots in Figure 1E are gated on CD4⁺PD-1⁺ and CD8⁺PD-1⁺ cells (as usually done in the literature in order to analyse the proportions of PE and TE cells with the additional TCF-1 and TIM-3 stainings), as indicated above

Revision Plan

*the plots on the left figure. In addition, the figure on the right present the percentages of TE T cells among total T cells. In order to increase the clarity of these 2 figures, we have separated them in **Figures 1E and F**, respectively, and modified the **legend of Figure 1**.*

Fig S2

As the authors only show FACS plots for the IFNARko cells, it is important to specify whether the statistical analysis and the graph representations in Fig S2A are based on the same data they show in Fig 1 or they performed parallel experiments with WT, WT-IFNARko, STING-GOF, STING-GOF-IFNARko mice.

*Yes, as mentionned by the referee, statistical analysis and the graph representations in previous Figure S2 (in the new version, Expanded View Figure 3A) are based on the same data showed in Figure 1 for STING GOF mice. This has been more precisely written in the **legend of Expanded View Figure 3**.*

Fig 2

NFATc1 levels are now not significant as in in Fig S3B. Is this due to number of experiments/mice? I would suggest to repeat so as to test if the result in S3B is reproducible and to also to show FACS plots.

*Thank you for this relevant comment. In order to reply to this point, we have done and added in the **Figure 2** additional mice (n=2), in order to have n=6 mice per group. With these new data, we show that NFATc1 is increased in STING GOF non treated T cells (red), as well as in WT DMXAA-treated T cells (light blue, compared to WT non treated T cells (black), in a statistically significant manner, after 24h of culture in vitro, which is more coherent with the ex vivo results (**Expanded View Figure 4B, which now replaces previous Figure S3**) and with our conclusions.*

Fig 3

- In Fig 3A the authors show reduced levels of CD3, which agrees with impaired in vitro responses with aCD3 in their previous study (Buis et al, 2018, JACI). The authors assume that this is due to "consistent TCR stimulation". However, there is no evidence for this. It is plausible that CD3 downregulation is a consequence of exhaustion or that there is less gene expression (their RNAseq analysis could answer this). Do exhausted T cells in the Rag1 hypomorphic model also have reduced CD3?

*As mentioned by the referee, the decrease of CD3 expression is indeed consistent with the significant decreased response of STING GOF T cells to anti-CD3 stimulation in vitro in our previous publication. We have added this point in the results part, **lines 256-258**. As explained before, STING GOF CD4⁺ and CD8⁺ naive T cells seem to be only moderately concerned by the exhausted phenotype. It allows us to hypothesize that exhaustion starts to set up after TCR (and also IL7R, see further points) activation, because the downregulation of expression of IL7R α and CD3 already takes place on naive T cells in these mice (new data added for CD3 expression in **Expanded View Figure 5**, and in the text of results section **lines 261-263**). In addition, CD3*

Revision Plan

*downregulation was also detected in Rag hypomorphic mice, which also display T cell exhaustion. These results have been added in a **new Expanded View Figure 7C**, with modified legend, and in the text of the results, **lines 355-357**.*

*Consistent with the internalization of the TCR complex upon its stimulation, the down regulation of CD3 expression was only detectable at the protein level, since mRNA levels were not diminished in our RNAseq analysis. This has been added in the text results section **lines 258-260** and in **Appendix Table S2**.*

*Finally, we showed a significant engagement of TCR in STING GOF T cells compared to T cells from control mice using four GSEA signatures. This was added in the text of the results **lines 253-254** and in **Appendix Table S1**. In addition, we would like to inform the referee that the GSEA TCR signature initially used in the manuscript in Fig. 3A (WP T Cell Antigen Receptor TCR Signaling Pathway) was not available anymore in GSEA application at the date of last consultation (January, 10th, 2025). Therefore we propose to remove it from the paper. Then we have replaced the signature in Figure 3A by the "PID-TCR-PATHWAY" signature.*

Altogether, these data strongly suggest that there is a consistent TCR stimulation in STING GOF T cells, which seems to precede the onset of T cell exhaustion.

Fig 3

- I am not sure I understand the data in Fig 3C. Isn't this simply because they are dividing with a smaller number, since STING GOF mice have much fewer cells? They need to show cDC numbers. Also, what does number of cDCs per CD4 T cell means? It is usually one cDC associating with multiple T cells, not the other way around. They could perform microscopy if necessary to investigate number of T cells associating per cDC or duration of cDC-T cell contacts, but I don't think the current analysis/data suggests enhanced TCR signaling. In this regard, I think the discussion points about the role of DCs need to be revised.

*We thank the referee for this question about cDCs numbers. Indeed, this increased ratio of cDC versus T cells in STING GOF mice is really linked to a decreased number of T cells. The numbers of splenic cDCs is not modified in STING GOF mice compared to control mice. These data (numbers of T cells and cDCs) have been added in the text of the results **lines 270-278**. The model we propose here is therefore the following: instead of saying that one T cell will associate with multiple cDCs, we propose that the very few STING GOF T cells have more chance to receive activation signal through TCR-MHC complex, mainly provided by cDC whose number is normal. The text has been modified to this effect (**lines 278-281**). We have also revised the discussion by removing the sentence "Of note, the observation that the lymphopenia leads to a higher number of cDCs per CD4⁺ T cells than for CD8⁺ T cells may help explain the more severe T cell exhaustion phenotype observed for CD4⁺ T cells" (**lines 402-404**). About the comment in enhanced TCR signaling, please the reply to the other comment above.*

- In Fig 3D the authors show a remarkable downregulation of IL-7Ra. Although IL-7 does rapidly downregulate IL-7Ra this is usually seen after in vitro stimulation with rather high doses of recombinant IL-7. Do these mice produce very high levels of IL-7, which could help explain the observed downregulation of IL-7Ra? Or is this also a consequence of exhaustion? In this regard,

Revision Plan

do T cells in the Rag1 hypomorphic model also have reduced IL-7Ra? Is IL-7Ra gene expression normal in STING-GOF T cells?

Besides homeostatic proliferation, IL-7 through STAT5 induces Bcl2 and survival while it is a potent inducer of surface lymphotoxin. Do STING-GOF T cells have high Bcl2 or lymphotoxin expression? This would show, indirectly at least, that there is active IL-7-IL-7R signaling.

Please find below our responses to these comments:

- *We also observed a significant decrease of mRNA encoding IL7R α in our RNA seq analysis in CD4⁺ and CD8⁺ T cells. We have added these data in the text of the results **lines 288-298** and in a **new Appendix Table S2**. Therefore, it is an additional data in favor of IL-7R activation. Indeed, according to the literature (see for example this review: <https://doi.org/10.1016/j.cyto.2022.156049> and in the references added in the results section **lines 294-295**), the engagement of IL-7R by IL-7 downregulates IL7R α expression both at the transcriptional and protein level, and both are seen on STING GOF T cells.*
- *Other data are also in accordance with the engagement of IL-7R in STING GOF T cells:*
 - *bcl2 mRNA expression is increased in CD4⁺ T cells in STING GOF mice, and also in STING GOF CD8⁺ T cells in a statistical manner.*
 - *The mRNA encoding p27^{kip1} (cdkn1b gene), a member of the universal cyclin-dependent kinase inhibitor (CDKI) family, is significantly decreased in STING GOF CD4⁺ and CD8⁺ T cells in our RNA-seq analysis. It is well described in the literature that this gene is known to be negatively regulated after IL-7R stimulation.*
 - *Another gene known to be regulated by IL-7R activation is gfi1, encoding GFI1 (growth-factor independent 1) which is induced by IL-7 stimulation to repress IL-7R α expression. In our RNA-seq analysis, we observed a significant decreased of gfi1 mRNA both in CD4⁺ and CD8⁺ T cells in STING GOF mice compared to control mice.*
 - *Finally, SOCS-1 (suppressor of the cytokine signaling 1) is another target gene of IL-7R engagement, which is upregulated after stimulation with IL-7, to negatively regulate IL-7R signaling pathways. We observed an increased expression of mRNA encoding SOCS-1 in STING GOF CD4⁺ and CD8⁺ T cells compared to control cells, in the RNA-seq analysis.*
 - *We added these data in the text of the results **lines 288-298**, with three references of reviews about IL-7 biology, and in the new **Appendix Table S2**.*
- *Concerning the potential causes of IL-7R increased stimulation in STING GOF T cells:*
 - *We previously did an experiment to measure serum IL-7 level in STING GOF mice, with ELISA kit (ref Invitrogen Mouse IL-7 ELISA kit EMIL7). Unfortunately, although the standard curve was correct, we didn't manage to detect any IL-7 in the serum, even in low-diluted serum (dilution 1/2), in control and STING GOF mice. The assay range is announced by the supplier as 8.23-6000 pg/mL. Because we didn't measure any detectable level of IL-7 in serum*

Revision Plan

of STING GOF mice, we can suppose that, if increased, is it not sufficiently increased to be detectable by ELISA kits. As mentioned in the literature on IL-7, serum IL-7 levels in mice are very low and difficult to detect by ELISA or immunochemistry (references 32-33). Indeed, the mean level of IL-7 in serum in normal mice is described in the literature as being less than 10 pg/ml, depending on the references (see for example Figure 5 in this reference: <http://www.biomedcentral.com/1471-2407/10/12>). In addition, it is described that the amount of IL-7 is regulated more by consumption than by production, and that the response of cells to IL-7 is more regulated by IL-R α expression than by the production of IL-7 itself (references 32-33).

- *We can hypothesize that T cells in the situation of T cell lymphopenia in STING GOF mice are more activable by IL-7 because they will encounter an abundant amount of IL-7 per T cell, compared to a physiological situation, as described in lymphopenia-induced proliferation. This has been added in the results section, **lines 295-298**.*

- *In addition, we perform new experiments on hypomorphic Rag mice to analyse the expression of IL7R α by flow cytometry. Our results showed that IL7R α expression is also reduced on CD4⁺ and CD8⁺ T cells from Rag1 hypomorphic mice (**new Expanded View Figure 7D**) and in the results section **lines 357-358**, reinforcing the hypothesis of lymphopenia being a major cause of T cell exhaustion.*

- *Finally, in STING GOF mice, IL7-R α expression is already decreased at the stage of naive T cells (**Expanded View Figure 5**) where exhaustion seems to be only moderate (**new Expanded View Figure 1**). Therefore, it allows us to propose that IL-7R stimulation, which is surely induced by lymphopenia, as the TCR stimulation, precedes exhaustion and is not only a consequence. The discussion part has been completed with this model proposal in **lines 407-414**.*

Discussion

- Lines 355-356: the authors state that cDCs are radioresistant. I am not sure that this is correct. In my experience, cDCs are easily "wiped out" by lethal irradiation and there are numerous papers using BM chimeras as a technique to test cDC-intrinsic biology. References or experimental evidence to substantiate this statement is important.

*Thanks to the referee for this comment. The referee is right, this was a mistake, we apologize for this. We have removed the following part of the sentence about cDC in the discussion **lines 434-435**: "which are also included in the radioresistant cell population".*

Minor comments:

Revision Plan

- Line 147: it is worth mentioning here that downregulation of AP-1 transcription factors is a hallmark of T cell exhaustion, at least in the context of viral infection (see Wherry et al, 2007, Immunity).

*We have modified the sentence in the results section **line 166** in order to strengthen this point.*

- In Fig 1, z-score scale bars are missing color code.

There must have been an error when formatting the manuscript. We hope that this is now corrected. In case the error is still present on your copy, please find below the color scale.

Z-score

- In Fig 2 definitely change the DMXAA color from orange to something more obvious next to the red STING-GPF (maybe blue?).

*We have changed the color of the orange points, to a light blue color, for the entire **Figure 2**.*

- Any potential mechanisms as to how STING activation induces Ca²⁺ signaling and NFAT activation? Could it be TBK1-IRF3 or TBK1-NFκB or IFNAR dependent?

*A negative regulation of STING by STIM1 has been described in the literature (Srikanth S and coll, Nat Immunol, 2019, 20(2), 152-162): in resting state in WT cells, STIM1 retains STING at the ER. When STING is activated (via cGAMP or SAVI gain of function mutations), the STING–STIM1 interaction is disrupted, which allows STING to exit the ER, moves to the ERGIC, and begin trafficking and signaling. We can therefore hypothesize that there could be a negative inter-regulation of STING and STIM1 at the ER: in case of STING activation (as in our STING GOF model or after DMXAA treatment in WT cells), STIM1 is not retained by STING at the ER anymore and could move to the ER-PM (plasma membrane) junction to operate SOCE (store-operated Ca²⁺ entry) and allow entry of calcium into the cell, then activating NFAT pathway. This hypothesis has been added in the text of the discussion **lines 423-428**.*

*In addition, the NFAT activation upon DMXAA treatment was already described to be type I IFN-independent, as it was already indicated in the discussion part **lines 421-423** (and reference Wu et al, 2020).*

- Related to Fig S3. In lines 200-201 the authors state that the data in Fig S3 suggest "that Ca²⁺-NFAT signaling pathway in STING GOF T cells participates to the induction of their exhaustion".

Revision Plan

However, this is only a correlation. It could be that these observations are the consequence of exhaustion and not the reason for exhaustion. As it stands, the statement cannot be substantiated and therefore I would suggest that the authors change it.

*We replaced “participates” by “could be associated with” **line 232** in the text of the results.*

- In their previous publication (Bouis et al, 2018, JACI) show that approximately 45% of STING-GOF mice get lung disease. It would be interesting if there was any correlation between the degree of exhaustion and disease severity.

*We have never analyzed T cell exhaustion in blood, lymphoid organs or lungs in parallel with lung disease in the same mice. However, while lung disease in mice can be heterogeneous (as the 45% of disease described in our previous paper), T cell exhaustion is present in all the mice, as can be seen in Figure 1. Thus, at this stage, we cannot conclude for a direct correlation between lower potential exhaustion and a more severe disease. In addition, other factors are surely implicated in the severity of the disease. We have added a sentence to this effect in the discussion **lines 380-382** for the sake of clarity.*

- I don't think the connection between TCR, IL-7Ra, lymphopenia and exhaustion is clear. Could the authors please elaborate potential mechanisms? Especially for IL-7Ra.

*We hope that the referee is more convinced with the new experiments added in the manuscript (see the reply to other points in this revision plan), and the sentences added in the new discussion part in this point **lines 407-414**.*

- Line 290: there is a "had" missing between "mice" and "similar".

We have corrected it.

Strengths and advances

The study is novel, the experiments thoroughly performed and I think it addresses a very important question: mechanisms and causes of T cell exhaustion. As the authors rightly state and cite, these data could have important future implications in hematopoietic cell transplantation but I think also **in patients receiving immunosuppression and potentially in CAR T therapy for T cell malignancies**. In addition, the mouse models could be invaluable for elucidating mechanisms of exhaustion or testing treatments. It is also very encouraging to see the correct statistical tests performed. It is true, and cited by the authors, that the association between lymphopenia and T cell exhaustion has been shown before. However, to the best of my knowledge, there are currently no models or mechanistic insights and I think the current study is a promising new advance.

*We have added this point (in red) in the discussion **line 456**, because we do agree with the referee*

Revision Plan

that it would not only concern hematopoietic cell transplantation (HCT) clinical situation, and deserves to be added in the discussion.

Limitations

I do not find any major weaknesses in this study, with the exception of some overinterpretations and some experiments that can be easily performed to substantiate the findings.

Audience

The study will be relevant for basic and clinical immunologists.

Expertise

My area of expertise is T cell biology. A key focus of my lab is checkpoint receptors in the context of cancer immunotherapy and signaling pathways that co-regulate development and inflammation.

Reviewer #2:

Using a mouse model bearing a gain of function mutation of STING (V154M) the authors report that, in a lymphopenia context, the remaining splenic and lung CD4 and CD8 T cells present an exhausted phenotype (by RNAseq and Flow cytometry) starting very early in life. The authors showed that this phenotype is IFN-Type I independent. Interestingly their results suggest that T cell exhaustion in this model might be a consequence of the lymphopenia. Accordingly, two other mouse models with lymphopenia present enrichment of T cells positive for exhaustion markers, albeit to a lesser level.

This is a well conducted study that took advantage of a mouse model developed and characterized by the team. The experiments are adequately replicated and all required controls were included. The statistical analyses are generally appropriate. It is well written and the figures are clear.

Major comments:

1) From RNAseq analysis and flow cytometry detection of surface markers, the authors conclude that the remaining splenic and lung CD4 and CD8 T cells in the STING GOF mouse model are exhausted. This fits with previous observation of reduced proliferation and increased cell death of T cells in this mouse model. A direct functional validation would however be an added value. The authors could quantify by flow cytometry the production of effector cytokines (eg. IL2, Granzyme, IFN-g) by T cells to confirm the increased frequency of exhausted T cells in their model. Moreover, if as proposed in the discussion, exhaustion is actually protective in the case of STING GOF, it would be pivotal to unravel whether phenotypically exhausted and non-exhausted T cells have distinct cytokine secretion abilities in this context.

Thanks to the referee for this comment.

Revision Plan

It is right that we already showed in our previous paper that proliferation rate of STING GOF T cells is decreased compared to WT control cells, and this somehow a functional defect.

Indeed, we already tried to analyze the production of some cytokines (as IFN γ) by intracellular staining in STING GOF T cells. However, we have encountered some limitations, as the induction of the STING pathway is known to strongly increase the production of IFN γ and induce TH1 pathway (see for example this paper: <https://doi-org.proxy.insermbiblio.inist.fr/10.1136/jitc-2021-003459>). Consequently, it is impossible to assess the functional impact of exhaustion in STING GOF T cells by comparing them to WT cells. Thus, we tried to compare IFN γ expression between exhausted and non exhausted STING GOF T cells to better appreciate functional defects. Because IFN γ staining is too low at basal state to draw any conclusions, we were obliged to stimulate T cells to increase cytokine production. However, after stimulation with anti-CD3 and anti-CD28, the immune checkpoint expression by STING GOF T cells is so high (almost 80-90% taking into account PD1 or TIM3 staining) that it is no longer possible to determine which cells were initially exhausted.

*Taking these considerations into account, we decided to perform a new experiment to address the referee's comment. We sorted exhausted and non exhausted CD4⁺ and CD8⁺ T cells from STING GOF mice by flow cytometry, and analyzed by qPCR the expression of genes encoding key effector cytokines (IFN γ , IL-2, granzyme B). We think this is the best way to reply to this question in the face of the increased production of IFN- γ induced by STING pathway. Our results showed that the expression levels of *Ifng* and *Gzmb* mRNA were significantly reduced in exhausted CD4⁺ and CD8⁺ T cells, in a significant manner. These results have been added in a **new Figure 1G and H**, and in the results section **lines 184-187**. Detection of IL-2 transcripts in this experiment were too low (high Ct) to be taken into consideration for the analysis.*

3) The authors observed a decreased surface expression of IL7Ra on total CD4 and CD8 T cells. It is further showed for naïve but not for the memory T cell subsets. As IL7 and IL15 are important for homeostatic proliferation of memory CD8 T cells (JT. Tan et al JEM 2002) IL7ra and IL15ra expression on memory T cells (by flow cytometry) should be assessed to support the author conclusion. Moreover, dosing both cytokines in serum and spleen supernatant (by ELISA for example) would support the proposed mechanism.

*We added the data for memory T cells: our data show that memory T cells in STING GOF mice display a high decrease of IL-7R α expression compared to T cells from control mice, as do naive T cells. These data were added in **Expanded View Figure 5** and modified the text in the results section **lines 285-288**.*

For the dosage of serum IL-7, please refer to the reply made to referee 1 in section 2, and also the other points about IL-7R pathway.

About the comment on IL-15, please refer to section 3.

Minor comments:

Revision Plan

- Although the model has already been described it will help the reader to show the total number of CD4 and CD8 T cells (figure 1, S1, 5 and S5), particularly if the cDC/T cell ratio is one of the driving force behind exhaustion.

Thanks to the referee for suggesting to mention the number of CD4⁺ and CD8⁺ T cells in STING GOF mice. These have been added to the text of Results section in relation to cDCs numbers, lines 274-278. See also the reply made to another comment raised by referee 1.

- Figure 1: the color code for the z-score is missing

As mentioned in this section for the minor comments of referee 1, there must have been an error when formatting the manuscript. We hope that this is now corrected. In case the error is still present on your copy, please find below the color scale.

- Figure 1 and 3: the FDR should also be indicated for the GSEA

FDR have been added for all GSEA signatures presented.

- Figure legends: the actual number of independent experiments should be indicated rather than "multiple".

The precise number of independent experiments have been added in each legend. In addition, the number of analyses has been updated for Figure 2 which incorporates new data in the revised version of the manuscript.

Reviewer #2 (Significance):

This study is conceptually important as it suggests that lymphopenia may promote and aggravate T cell exhaustion. The mechanisms at play are not clear but may be related to the availability of cytokines like IL7 and enhanced T cell activation through increased cDC/T cell ratio. These results may also have clinical consequences for the standard of care of patients presenting STING GOF mutations and more largely presenting severe lymphopenia. Indeed, T cell exhaustion might be an aggravating feature in case of cancer development in these patients.

Revision Plan

This study is likely to be of interest for clinical immunologists and scientists working in the fields of immunodeficiency and cell exhaustion.

Please note that we have moved up Grégoire Hopsomer's place in the list of authors to 3rd author, in view of his involvement in the experiments carried out to revise the manuscript.

3. Description of analyses that authors prefer not to carry out

Please include a point-by-point response explaining why some of the requested data or additional analyses might not be necessary or cannot be provided within the scope of a revision. This can be due to time or resource limitations or in case of disagreement about the necessity of such additional data given the scope of the study. Please leave empty if not applicable.

Reviewer #2 (continued)

Major comments:

2) Treating WT cells with an agonist of STING reproduced the increased basal Ca²⁺ level and upregulation of NFATc1 observed in STING GOF bearing T cells. However, this treatment had no effect on the expression of PD-1 and TIM-3. The authors conclude that STING activation is thus not sufficient to induce exhaustion. Although the chimera experiments indeed argue for an environmental driver for T cell exhaustion in this model, it can be argued that 24h of stimulation is not long enough to induce sufficient chronic activation to lead to exhaustion. This conclusion should either be dampened or a longer treatment with the STING agonist should be performed.

*We thank the referee for this remark. In fact, we tried to do some in vitro cultures of T cells from WT mice with DMXAA for 48h (and even 72h), and the mortality of the cells was very high. You can see the results of viability in the figure below for 24h of culture. It is coherent with the literature where it has been described that STING agonist induce a high apoptosis of cells in vitro. Therefore it won't be possible to do the experiment proposed by the referee. We added a sentence in the results section **lines 240-241** to explain this and dampen the conclusion.*

Revision Plan

3) The authors observed a decreased surface expression of IL7Ra on total CD4 and CD8 T cells. It is further showed for naïve but not for the memory T cell subsets. As IL7 and IL15 are important for homeostatic proliferation of memory CD8 T cells (JT. Tan et al JEM 2002) IL7ra and IL15ra expression on memory T cells (by flow cytometry) should be assessed to support the author conclusion. Moreover, dosing both cytokines in serum and spleen supernatant (by ELISA for example) would support the proposed mechanism.

About the remarks on IL-15:

As suggested and explained in section 2, we analyzed the expression of the IL-7R α on memory T-cell populations. These data show that the decrease of IL7-R α expression concerns both naïve and memory STING GOF T cells, compared to their WT controls, thus reinforcing the mechanism of homeostatic proliferation.

We do agree that, similar to IL-7, IL-15 plays a significant role in the homeostatic proliferation, especially for memory T cells. Coherently, our RNA-seq analysis confirms an IL-15 signature (REACTOME-INTERLEUKIN_15_SIGNALING) significantly enriched in STING GOF CD4⁺ (NES=1.85, p<0.001, FDR=0.007) and CD8⁺ (NES=1.56, p=0.014, FDR=0.071) T cells, compared to control T cells. It seems therefore that IL-15R pathway is also engaged.

Although dosing cytokines in serum is of interest, we prefer not to include this kind of analysis for the following reasons. First, these cytokines are so low in serum that quantification is not always possible. As explained in section 2, we didn't manage to detect any IL-7 in the serum of our mice because of its too low concentration (references 32-33). It will be either difficult to use IL-15 serum levels as a marker for homeostatic proliferation for the same reason. Indeed, IL-15 is almost undetectable in biologic fluids (see for example this paper:

[https://doi-org.proxy.insermbiblio.inist.fr/10.1016/s1074-7613\(02\)00429-6](https://doi-org.proxy.insermbiblio.inist.fr/10.1016/s1074-7613(02)00429-6)). As an example, Anderson and coll showed that the mean level of IL-15 in B6 mice is around 40 pg/ml while the detection threshold of murine IL-15 dosage ELISA kit is 125 pg/ml (<https://doi.org/10.1210/en.2015-1746>).

In addition to these technical issues, we believe that these cytokine concentrations, although indicative, are not necessarily representative of T cell stimulation. For example, IL-7 levels are directly regulated more by consumption than by production. In addition, IL-15 seems to mostly operate in a cell contact-dependent manner through the trans-presentation of membrane-bound complexes of IL-15 and IL-15R α on the producing cells to IL-2/IL-15 receptor- β chain (CD122) and γc on the responding cells (<https://doi-org.proxy.insermbiblio.inist.fr/10.4110/in.2024.24.e11>), suggesting that the soluble IL-15 doesn't seem to be the best correlate with potential IL-15R engagement.

Our primary focus in Figure 3, is on the impact of lymphopenia and homeostatic proliferation on naïve T cells, since it appears that these cells are already impacted by TCR and IL-7 stimulation before acquiring the T cell exhaustion phenotype. Therefore, we would prefer to concentrate here on IL-7 since naïve T cells are highly dependent on IL-7 for their survival and homeostatic maintenance, particularly under conditions of lymphopenia. While IL-15 plays an important role in the proliferation and persistence of memory T cells, introducing this additional layer of complexity

Revision Plan

is less aligned with our current objectives. Focusing exclusively on IL-7 allows us to better understand the specific dynamics affecting naive T cells in STING GOF mice without complicating the analysis with homeostatic proliferation of memory T cell, which also seems to occur in our mice (GSEA signature of IL-15R engagement can be added in supplemental if suggested by the referee) but which we believe is less connected with T cell exhaustion acquisition.

1st Jul 2025

Dear Prof. Soulas-Sprauel,

Thank you for submitting your revised study, and please accept my apologies for the delay in getting back to you as I was traveling for work, and we also received a high number of submissions recently. Referees #1 and #2 reviewed your revised manuscript and as you will see below, they are satisfied with the revisions. I will therefore be able to accept your manuscript once the following editorial concerns are addressed:

1/ Referees' concerns:

Please address the remaining minor concerns from referee #2.

2/ Manuscript text:

- Please remove the red font text and only keep in track changes mode any new modification in the text.
- Remove: Classification: Biological Sciences - Immunology and Inflammation.
- Please note that emails for Marita Bosticardo (bosticardo.marita@hsr.it) and Frederic Rieux-Laucat (rieux@necker.fr) bounced. Kindly check and correct as appropriate.
- Methods:
 - o Mice: please provide the strain and age of the mice.
 - o Antibodies: provide dilutions/concentrations for all antibodies.
 - o Cells: indicate whether the cells were tested for mycoplasma contamination.
 - o Statistics: please provide statements on sample size, inclusion/exclusion criteria, blinding and randomization.
 - o Please clarify "Taking into account that data didn't follow a normal distribution OR the samples size is under xxx".
- Data availability: please provide a URL to the deposited RNAseq data.
- Move the "Adherence to community standards" section to the Methods.
- Acknowledgements: please make sure that the complete list of funders that need to be mentioned is entered into our system. Currently, a number of funders are only listed in the Acknowledgements.
- Please provide a Disclosure statement and competing interests statement: Please review the policy <https://www.embopress.org/competing-interests> and update your competing interests if necessary.

3/ Figures:

- Appendix: the nomenclature in the legends should be corrected to "Appendix Table S1 and S2".
- Figure callouts: please add panels A and B in the callouts for Fig. EV3.
- Please address the queries from our data editors in the figure legends:
 1. Please note that the exact p values are not provided in the legends of figures 1A, D, F, H; 2B, C; 2A-D; 4B, E; 5, EV1 A, B; EV2 A-C; EV3 A, B; EV4 B, EV5 B, C; EV6 B, E; EV7 A-D.
 2. Please indicate the statistical test used for data analysis in the legend of figure EV4 A.

4/ Source Data:

The file for Source Data for figure 5 appears empty on BioStudies. Please check and correct.

5/ Checklist:

- Cells: please indicate whether the cells were tested for mycoplasma contamination.
- Statistics: please fill in the entire section on Experimental study design and statistics.

6/ Synopsis:

I introduced minor changes in your synopsis text, please let me know if you agree or amend as you see fit:

"Lymphopenia was identified as the key driver of peripheral T cell exhaustion in a murine model of SAVI (STING-Associated Vasculopathy with Onset in Infancy). These results have implications for the management of patients with severe lymphopenia.

- A terminal exhaustion phenotype was observed in CD4⁺ and CD8⁺ T cells from STING V154M mice.
- Exhaustion occurred independently of type I interferon signaling or STING activation in T cells or stromal cells.
- Increased IL-7R, TCR and NFAT pathway activity was detected in naive T cells of STING GOF mice.
- Exhaustion of STING GOF T cells was prevented in a non lymphopenic environment."

I have cropped a small portion of your visual abstract to serve as thumbnail for the table of content on our webpage (attached). Please let me know if you agree, or provide an alternative image at the right dimensions (115px x 70px).

7/ As part of the EMBO Publications transparent editorial process initiative (see our Editorial at <http://embomolmed.embopress.org/content/2/9/329>), EMBO Molecular Medicine will publish online a Review Process File (RPF) to accompany accepted manuscripts.

This file will be published in conjunction with your paper and will include the anonymous referee reports, your point-by-point response and all pertinent correspondence relating to the manuscript. Let us know whether you agree with the publication of the RPF and as here, if you want to remove or not any figures from it prior to publication. Please note that the Authors checklist will be published at the end of the RPF.

I look forward to receiving your revised manuscript.

Yours sincerely,

Lise Roth

***** Reviewer's comments *****

Referee #1 (Comments on Novelty/Model System for Author):

This is a well conducted study that took advantage of a mouse model developed and characterized by the team. The experiments are adequately replicated and all required controls were included. The statistical analyses are generally appropriate. It is well written and the figures are clear. This study is conceptually important as it suggests that lymphopenia may promote and aggravate T cell exhaustion. These results may also have clinical consequences for the standard of care of patients presenting STING GOF mutations and more largely presenting severe lymphopenia. Indeed, T cell exhaustion might be an aggravating feature in case of cancer development in these patients.

Referee #1 (Remarks for Author):

In this revised manuscript the authors describe lymphopenia as a cause for T cell exhaustion. They provide further evidence supporting a role for IL7 signaling in this process and they performed complementary experiment confirming functional exhaustion in addition to expression of exhaustion markers. They thus addressed most of my comments.

They could not detect IL7 and IL15 for technical issues. Neither could they assess T cells treated with a STING agonist for longer than 24hours due to high mortality. This was taken into account in their conclusion and in their discussion and is acceptable as is.

Referee #2 (Comments on Novelty/Model System for Author):

I think this work is of interest to all immunologists and medical scientists studying T cell exhaustion. Perhaps not other disciplines.

I think that after the corrections the paper is suitable for publication.

Referee #2 (Remarks for Author):

The authors have adequately addressed all my comments.
I only have two points:

- 1) In my comment regarding DC/T ratio analysis I would suggest to show cDC and T cell numbers in the figure next the DC/T ratio graphs; it is more visual than stating statistics in the text
- 2) I would suggest that in line 299 the authors change "suppose" to "suggest" and "activable" to "activated"

Rev_Com_number: RC-2024-02769

New_manu_number: EMM-2025-21368-V2

Corr_author: Soulas-Sprauel

Title: Lymphopenia drives T cell exhaustion in immunodeficient STING gain-of-function mice

The authors addressed the remaining editorial issues.

25th Jul 2025

Dear Prof. Soulas-Sprauel,

Thank you for submitting your revised files. I am pleased to inform you that your manuscript is accepted for publication and is now being sent to our publisher to be included in the next available issue of EMBO Molecular Medicine!

Please note that I have included the following sentence in your Data Availability section:

Source Data for Figures 1, 3 and 5 have been deposited on BioStudies: <https://www.ebi.ac.uk/biostudies/studies/S-BSST2071?key=bf3ffafd-8eef-45e9-bfde-15a557d8e3d0>

Let us know immediately if you have any objection.

Yours sincerely,

Lise Roth

Rev_Com_number: RC-2024-02769

New_manu_number: EMM-2025-21368-V3

Corr_author: Soulas-Sprauel

Title: Lymphopenia drives T cell exhaustion in immunodeficient STING gain-of-function mice